



# Generation of global 1 km daily land surface - air temperature difference and sensible heat flux products from 2000 to 2020

Hui Liang[1], Shunlin Liang[2], Bo Jiang[3,4], Tao He[1], Feng Tian[1], Jianglei Xu[2], Wenyuan

Li[2], Fengjiao Zhang[1], Husheng Fang[1]

[1]Hubei Key Laboratory of Quantitative Remote Sensing of Land and Atmosphere, School of Remote Sensing and Information Engineering, Wuhan University, Wuhan, 430079, China

[2]Jockey Club STEM Lab of Quantitative Remote Sensing, Department of Geography, University of Hong Kong, 999077, China

[3]State Key Laboratory of Remote Sensing Science, Faculty of Geographical Science, Beijing Normal University, Beijing 100875, China

[4]Beijing Engineering Research Center for Global Land Remote Sensing Products, Faculty of Geographical Science, Beijing Normal University, Beijing 100875, China

*Correspondence to*: Shunlin Liang (shunlin@hku.hk)

**Abstract.** Accurate estimation of land surface sensible heat flux (H) is crucial for comprehending the dynamics of surface energy transfer and the cycles of water and carbon. Yet, existing H products mainly are meteorological reanalysis datasets with coarse spatial resolutions and high uncertainties. FLUXCOM is the sole remotely sensed product with its 0.0833° spatial and 8-day temporal resolution spanning from 2001 to 2015, so there is still a need for accurate and high spatial resolution global product based on

satellite data. To address these issues, we generated the first global high resolution (1km) daily H product from 2000 to 2020 using long short-term memory (LSTM) deep learning models, incorporating data from the Global LAnd Surface Satellite (GLASS) product suite. Furthermore, considering that the difference between land surface temperature and air temperature (Tsa) is a key driver of H, we introduce the first global accurate satellite-based Tsa product. This product refines the uncertainty compared with obtaining

Tsa directly from existing products by subtracting air temperature from land surface temperature. Our model, distinct from previous models that estimate H per pixel through physically-based models requiring parameters that are not readily accessible, can conveniently derive global values and efficiently capture nonlinear interactions. Additionally, it accounts for the temporal variation of H. Validation against independent in-situ measurements yielded a root mean square error (RMSE), mean absolute error (MAE),

and coefficient of determination ($R^2$) of 25.54 Wm$^{-2}$, 18.649 Wm$^{-2}$, and 0.54 for H, and 1.459 K, 1.071

K, and 0.53 for Tsa, respectively. The estimated H and Tsa values are more accurate than current products such as MERRA2, ERA5-Land, ERA5, and FLUXCOM under most conditions. Additionally, the new H product offers more detailed spatial information in diverse landscapes. The estimated global average land surface H from 2000 to 2020 is $35.29\pm0.71$ Wm$^{-2}$. These high-resolution H and Tsa products are
invaluable for climatic researches and numerous other applications. The daily mean values for the first three days of each year can be freely downloaded from https://doi.org/10.5281/zenodo.14986255 (Liang et al., 2025), and the complete product will be available at www.glass.hku.hk (last access: 7 March 2025).

## 1 Introduction

Sensible heat flux (H) is the turbulent transfer of heat between the land surface and the atmosphere,
primarily driven by temperature differences ($T_0$–$T_a$, also referred to Tsa, where $T_0$ (K) represents the surface aerodynamic temperature and $T_a$ (K) the air temperature) (Mito et al., 2012). As a major energy source for the lower atmosphere and a critical component of surface energy fluxes, H plays a vital role in land-atmosphere interactions, particularly in thermal exchanges between the land surface and the atmospheric boundary layer (Beamesderfer et al., 2023; Liao et al., 2019). The uneven distribution of H
leads to alternating absorption or release of heat into the atmosphere, affecting monsoon circulation and local climate systems (Mito et al., 2012; Zhou and Huang, 2014; Zhou and Huang, 2010). Therefore, accurate estimation of H is essential for studying global energy flows and understanding the dynamic transfers of water, energy, and trace gases at the Earth's surface (Watts et al., 1997).

Currently, H can be derived from ground-based measurements or various products. Ground
measurements, considered as "ground truth" values, are typically obtained using eddy covariance (EC) system (on the scale of hundreds of meters) and large aperture scintillometer (LAS, at the kilometer scale). The EC system measures instantaneous variations in vertical wind speed and scalar quantities (e.g. temperature, carbon dioxide concentration, and water vapor), determining H by calculating the covariance between these variables (Zhang, 2024). In contrast, LAS estimates H using the scintillation
principle, measuring light signal disturbances caused by atmospheric turbulence (Liu et al., 2011). These instruments have shown reliable performance across scales from tens of meters to approximately 1 km, with reported differences of 2–3% in EC and LAS measurements (Liu et al., 2018; Liu et al., 2011; Baldocchi et al., 2001). However, their practical applications are limited to areas with flat, uniform terrain

and stable turbulent conditions, leading to shorter observation periods and sparse spatial coverage due to high maintenance costs (Mito et al., 2012). Another alternative for obtaining H is through existing products, including reanalysis products generated by merging available observations with land surface models and remotely sensed products derived from satellite data via machine learning techniques. Table 1 lists the mainstream products utilized for analysis and evaluation across various global applications,

including the Interim Reanalysis (ERA-Interim) and its latest version ERA5 and ERA5-Land from the European Centre for Medium-Range Weather Forecast (ECMWF), the Japanese 55-year Reanalysis (JRA-55), Climate Forecast System Reanalysis (CFSR) from the National Centers for Environmental Prediction (NCEP), the Modern-Era Retrospective analysis for Research and Applications Version2 (MERRA2), The Global Land Data Assimilation System (GLDAS) and FLUXCOM. These products generally provide long temporal coverage but tend to have coarse spatial resolution and significant

uncertainty, as illustrated in Table 1. Even FLUXCOM_RS, the most recent and only satellite product boasting the highest spatial resolution of 0.0833°, encounters a global uncertainty of 11.61% over an 8-day period from 2001 to 2015, thereby restricting its utility for local-level applications. Products that combine high precision with finer resolution are crucial, particularly for supporting studies in regions with complex land cover and climate features, such as the Tibetan Plateau (Yizhe et al., 2019) and urban

areas with complex human activities (Kato and Yamaguchi, 2005).

**Table 1.** The mainstream global product information.

| Product | Period | Resolution | Uncertainty | Method | Reference |
|---|---|---|---|---|---|
| Reanalysis data products | | | | | |
| ERA5-Land | 1950-present | 0.1° × 0.1° 1 hourly | RMSE=38.21 Wm⁻² in TP at daily scale | ECMWF land surface model | (Xin et al., 2022; Muñoz-Sabater et al., 2021) |
| ERA5 | 1950-present | 0.25° × 0.25° 1 hourly | Similar to ERA5-Land | ECMWF land surface model | (Hersbach et al., 2020) |
| ERA-Interim | 1979-2019 | T255 (80 km) 3 hourly | RMSE=114.46 Wm⁻² in TP at daily scale | ECMWF IFS (Cy31r2) | (Xin et al., 2022; Berrisford, 2011; Dee et al., 2011) |
| JRA55 | 1958-present | T319 (~55 km) 3 hourly | \ | Similar algorithm as Beljaars et al. (Beljaars, 1995) | (Kobayashi et al., 2015) |
| CFSR | 1979-2010 | T382 (38 km) 6 hourly | RMSE=30–70 Wm⁻² at monthly | UA algorithm | (Decker et al., 2012) |





| | | scale | | | |
|---|---|---|---|---|---|
| MERRA2 | 1979-present | 1hourly 0.5° × 0.625° | \ | Updated Goddard Earth Observing System model | (Buchard et al., 2017) |
| GLDAS | 1948-present | 3 hourly 0.25° × 0.25° | \ | Data assimilation | (Rodell et al., 2004) |
| Remotely sensed products | | | | | |
| FLUXCOM_RS /FLUXCOM_RS + METEO | 2001-2015 | 0.0833°× 0.0833°/0.5 °×0.5° 8daily/daily | 11.61%/11.85% (1.59%/2.69% for Rn) in global at daily scale | Model tree ensembles | (Jung et al., 2019) |

H estimation has traditionally relied on temperature-derived one-source and two-source models, incorporating ground-based observations of temperature and wind fields. One-source models, treating the land surface as "one-leaf", have been extensively applied across various field crops at regional scales

over the past decade (Hatfield et al., 1984; Seguin et al., 1982a, b). Two-source models attempt to mitigate this by distinguishing between soil ($T_s$) and canopy ($T_c$) temperatures, yet they frequently overlook the role of precipitation interception in influencing energy distribution and surface temperature dynamics (Anderson et al., 1997; Anderson et al., 2007; Colaizzi et al., 2014; Kustas and Norman, 1999; Asdak et al., 1998). Both models face common challenges in calculating aerodynamic resistance ($r_{ah}$) due to the

complexities of Monin-Obukhov similarity theory (Monin and Obukhov, 1954; Brutsaert, 2013), and in accurately representing $T_0$ under diverse conditions, leading to significant uncertainties. Despite attempts to use the more easily obtainable land surface temperature (LST) as a proxy for its linear correlation with $T_0$ (Chehbouni et al., 2001), LST-related errors account for over half of the inaccuracies in these models (Timmermans et al., 2007; Stewart et al., 1994). Furthermore, the uncertainty associated with LST usage

could be up to four times higher than that of simulating all-wave net radiation (Rn) and ground heat flux (G) (Costa-Filho et al., 2021). Since Tsa significantly influences H, its variability directly reflects in H fluctuations. Therefore, improving the accuracy of Tsa estimation and minimizing related errors is crucial for developing a reliable, globally applicable method for H estimation.

Recent research has highlighted the significance of the Tsa, examining its influencing factors and

mechanisms of variation. Bartlett et al. (2006) found that downward shortwave radiation (DSR) is the primary factor affecting Tsa, with an observed increase of 1.21 K for every additional 100 $wm^{-2}$ of DSR. The absorbed DSR warms the land surface, influences Ta, alters H and enhances surface evaporation



(known as latent heat flux, hereinafter LE). In areas with dense vegetation, heightened evapotranspiration generally leads to evaporative cooling, which tends to reduce Tsa (Prakash et al., 2018; Gordon et al., 2005). Furthermore, Feng et al. (2019) reported that although albedo shows a weaker positive correlation with Tsa compared to LST and Ta, the normalized difference vegetation index (NDVI) exhibits a stronger correlation with Tsa than albedo or atmospheric water vapor. Additionally, factors such as terrain features (elevation and slope), snow cover, and precipitation have also been identified as influencers of Tsa (Cermak and Bodri, 2016; Jiang et al., 2022; Sun et al., 2020). Collectively, these studies indicate that Tsa is subject to a complex interplay of atmospheric and surface elements, complicating the attribution of its variability to any single factor (Feng and Zou, 2019). Moreover, most previous research has been carried out on regional scales, relying on in-situ measurements and reanalysis or remote sensing products (Liao et al., 2019), potentially leading to discrepancies in scale. Additionally, estimating Tsa by subtracting Ta from LST, using the same or different products, can introduce significant uncertainties (Wang et al., 2020).

Traditional physically-based model to get H mainly established in specific area and land surface condition with not easily accessible parameters (e.g. aerodynamic resistance to heat transfer, $r_{ah}$), it will have large uncertainties when applied in other areas. Therefore, there not have a suitable method for getting global values conveniently. Different from physically-base model, data-derive machine learning (ML) method has emerged as a formidable tool for enhancing the estimation of land surface parameters when adequate input data was adopted (Xu et al., 2022; Li et al., 2022b). Its superior performance and improved generalization capability position it as a potential solution for improving the accuracy and spatial resolution of Tsa and H on a global scale. Given the intricate interactions between Tsa and other land-atmosphere parameters, along with the significant temporal variations of H identified through our analysis, we utilized two machine learning methods, Random Forest (RF) and long short-term memory (LSTM), to predict Tsa and H, respectively. Initially, we employed the RF method, utilizing pertinent parameters mentioned above to precisely estimate Tsa, followed by an in-depth uncertainty analysis. Subsequently, a global H product for the period of 2000 to 2020 was generated using LSTM models, incorporating data from the Global LAnd Surface Satellite (GLASS) product suite and estimated Tsa. The remainder of this paper is organized as follows: Data and methodologies are detailed in Sections 2 and 3, validation results for Tsa and H are examined in Section 4, and the study's discussions and

conclusions are articulated in Sections 5 and 6.

## 2 Dataset and Pre-processing

This study utilized three distinct types of data: in-situ measurements, remotely sensed products, and reanalysis datasets. In-situ measurements were employed for both model development and independent validation. Remotely sensed products, including GMTED2010 DEM and GLASS product suite, supported the modeling and generation of new Tsa and H products, while FLUXCOM was used for comparison with H estimates. Reanalysis datasets were used for comparative analysis with H and Tsa estimates. Detailed descriptions of each dataset are provided in the subsequent sections.

### 2.1 In-situ measurements


    In this study, data from 398 sites across eight observation networks were collected for the period 2000–2019. The spatial distribution of these sites is shown in Fig.1 (a). The sites were globally distributed, with elevations ranging from –4 m to 4104 m above sea level, and were predominantly located in the mid-to-low latitudes of the Northern Hemisphere. These sites represent diverse land cover types and ecosystem

conditions within various climatic zones. In this study, the land covers were categorized into ten major classes based on the International Geosphere-Biosphere Programme (IGBP): Barren land with sparse vegetation <BSV>, Cropland <CRO>, Mosaic of crops and natural vegetation <CVM>, Forest <FOR>, Grassland <GRA>, Ice and Snow <IAS>, Savannas <SAV>, Shrubland <SHR>, Tundra <TUN> and Wetland <WET>. All sites were used for studying the Tsa, while 140 sites, marked with red triangle

symbols in Fig.1 (a), were specifically used for estimating H. The proportions of these sites across the ten land cover types and five elevation ranges for both Tsa and H studies are presented in Fig.1 (b-c). Furthermore, investigations by Li et al. (2022a), Jiang et al.(2023), and Yin et al.(2023) indicated that land cover types within a 5 km radius of most sites exhibited a high degree of similarity or equivalence. Consequently, these sites provide strong spatial representativeness and comprehensiveness.

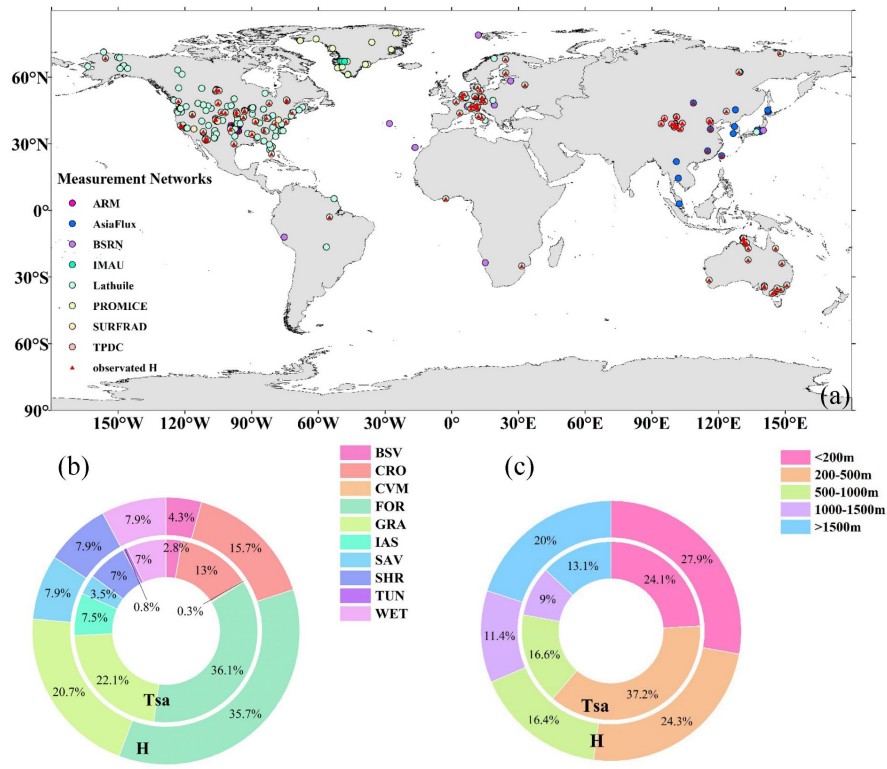

**Figure 1**. (a) Spatial distribution of 398 sites from eight measurement networks. The proportions of all sites for (b) ten land cover types; and (c) five elevation ranges.

Table 2 provides detailed descriptions of the eight observation networks encompassing the 398 sites, including Atmospheric Radiation Measurement (ARM), AsiaFlux, Baseline Surface Radiation Network (BSRN), the Institute for Marine and Atmospheric Research (IMAU), Lathuile (including FLUXNET and AmeriFlux), PROMICE, SURFRAD and National Tibetan Plateau/Third Pole Environment Data Center (TPDC). Selection of sites was predicated on the availability of specific measurements: those with $T_a$, downward longwave radiation (DLW), and upward longwave radiation (ULW) were utilized for Tsa calculation, whereas sites with LE, G, all four radiation components (DLW, ULW, DSR, and upward shortwave radiation <USR>) or Rn were used for H estimation. As indicated in Table 2, measurement data varied in frequency and format across the sites, necessitating the conversion of all quality-controlled measurements to local time. Subsequently, different methodologies were employed to calculate daily Tsa and H values. Tsa was determined at an instantaneous scale using $T_a$, DLW, and ULW measurements

according to Eq. (1). For sites recording data less frequently than hourly, Tsa data were compiled into

hourly averages, tolerating a maximum of 10 minutes of missing data per hour. These hourly values were

then aggregated into daily means with no missing data. Conversely, daily H values were computed

directly from days with over 75% of valid observations, and subsequently adjusted for energy imbalance

using the method proposed by Twine et al. (2000) (Eq. (2)). Rn measurements were acquired directly

from the sites or derived by summing the radiative components (Eq. (2a)). All calculated daily values

underwent manual verification to eliminate any implausible figures. The equations below detail the

procedures for calculating instantaneous Tsa and adjusting daily H values:

$$Tsa = (\frac{ULW-(1-\varepsilon)\times DLW}{\sigma\times\varepsilon})^{1/4} - T_a \qquad (1)$$

Where ε represent the surface broadband emissivity and σ is the Stefan– Boltzmann constant

$(5.67\times10^{-8}$ Wm$^{-2}$ K$^{-4})$.

$$H_{cor} = \frac{R_n-G}{H_{uncor}+LE_{uncor}} \times H_{uncor} \qquad (2)$$

$$Rn = DSR - USR + DLW - ULW \qquad (2a)$$

Where $H_{cor}$ is corrected H; $LE_{uncor}$ and $H_{uncor}$ are uncorrected LE and H, respectively.

**Table 2.** The detailed information about eight observation networks

| Abbreviation | No. of sites | Time Span | Instrument | Temporal resolution |
|---|---|---|---|---|
| ARM | 33 | 2000-2019 | Kipp&Zonen Pyrgeometer | 1 min |
| AsiaFlux | 21 (5) | 2000-2013, 2015-2018 | Kipp&Zonen CNR-1 | 30 min |
| BSRN | 20 | 2000-2019 | Eppley, PIR/Kipp&Zonen CG4 | 1 or 3min |
| IMAU | 3 | 2016-2019 | Kipp & Zonen, CNR-1 | 30min |
| Lathuile | 267 (116) | 2000-2019 | Kipp&ZonenCNR-1,etc | 30min |
| PROMICE | 27 | 2007-2019 | Kipp & Zonen CNR-1/CNR-2 | 1 hour |
| SURFRAD | 7 | 2000-2019 | Eppley, PIR | 3min |
| TPDC | 20 (19) | 2008-2010, 2012-2019 | CNR-4 | 10min |

Ultimately, a total of 649,895 daily Tsa measurements and 216,542 daily H in-situ measurements were

collected from 2000 to 2019 to estimate Tsa and H on a global scale. Due to significant gaps in the daily

in-situ measurements of H after stringent quality control, a distinct strategy was implemented to segregate

the samples for H and Tsa. For H, the methodology involved selecting monthly datasets with fewer than

10% missing values for the training set, while the rest were allocated to an independent validation set.

Linear interpolation was employed to impute missing values within the training set, ensuring the integrity

of the monthly datasets. A five-fold cross-validation was then applied, partitioning the data such that 80%

of the months were designated for training and the remaining 20% for testing during each iteration. This

process yielded a training set encompassing 121,542 daily H samples and an independent validation set

containing 97,982 samples. In contrast, the Tsa analysis designated measurements from 2018 to 2019 as

the independent validation set, with data from preceding years allocated to the training set. Specifically,

for each site, 70% of the 2000 to 2017 samples were randomly selected for the training set, and the

remaining 30% for testing. As a result, the Tsa training set included 564,918 daily samples, and the

independent validation set comprised 84,977 daily Tsa samples.

### 2.2 Remotely sensed data

#### 2.2.1 GMTED2010 DEM

The Global Multi-resolution Terrain Elevation Data 2010 elevation dataset (GMTED2010,

https://www.usgs.gov/coastal-changes-and-impacts/gmted2010, last access: 7 March 2025), developed

by the United States Geological Survey (USGS), provides an advanced level of detail in global

topographic data (Danielson and Gesch, 2011). It replaces Global 30 Arc-Second Elevation (GTOPO30)

as the preferred choice for global and continental-scale applications. GMTED2010 is produced by

combining multiple high-quality digital elevation model (DEM) datasets from various international

institutions. This dataset offers seven raster elevation products across three spatial resolutions: 30-, 15-,

and 7.5-arc-second. In this study, the 30 arc-second resolution product, which spans from 84°N to 90°S,

was employed to derive terrain attributes such as elevation, slope, and aspect. Detailed methodologies

for calculating slope and aspect are documented in the study of Liang et al (2023).

#### 2.2.2 GLASS product suite

The GLASS product suite (www.glass.hku.hk, last access: 7 March 2025) provides approximately

20 land surface variables with high spatial resolution (up to 250m) and long-term temporal coverage,

with many products extending from 1981 to the present. These products have gained widespread use in

land surface studies, attributed to their data integrity (no missing data) and superior quality (Liang et al.,

2021). The accuracy of GLASS products has been corroborated through numerous validations against

in-situ measurements and other existing datasets (Yin et al., 2023; Xie et al., 2022; Yang et al., 2023). In this study, eleven GLASS products, covering the period from 2000 to 2020, were selected based on their documented impact on H variations (Zhuang et al., 2016; Trenberth et al., 2009). These products are predominantly sourced from Advanced Very High Resolution Radiometer (AVHRR) and Moderate Resolution Imaging Spectroradiometer (MODIS) observations, supplemented by other satellite and ancillary data. Comprehensive details on these eleven products are provided in Table 3. The BBE product was specifically used to acquire in-situ LST measurements, and the $T_a$ and LST products were integrated to get GLASS Tsa for comparison with the estimated daily Tsa. Additionally, eight other products—surface broadband albedo (ABD), DLW, DSR, Rn, ET, FVC, NDVI and LAI—were employed to identify the optimal parameters for estimating H and to generate daily Tsa and H estimates. Specifically, LST, DLW, DSR, NDVI were used as model inputs for Tsa estimation, while the estimated Tsa, in conjunction with ABD, DLW, FVC, Rn, and ET, facilitated the estimation of daily H. To maintain spatial consistency, the DSR, DLW, and Rn products, originally at a 0.05° spatial resolution, and the NDVI and LAI products, at 250 m resolution, were resampled to 1 km using the bilinear interpolation method. Subsequently, values from these products were extracted according to the in-situ measurement locations.

**Table 3.** Summary of eleven GLASS series products used in this study

| Variables | Spatial resolution | Temporal Resolution | Usage | Reference |
|---|---|---|---|---|
| Land surface temperature (LST) | 1km | daily | Model input, comparison | (Li et al., 2024) |
| Air temperature (Ta) | 1km | daily | Comparison | (Chen et al., 2021) |
| Surface Broadband Albedo (ABD) | 1km | 8 day | Model input | (Qu et al., 2014) |
| Broadband Emissivity (BBE) | 1km | 8 day | Calculate in situ Ts | (Cheng et al., 2016; Cheng and Liang, 2013) |
| Downward longwave radiation (DLW) | 0.05° | daily | Model input | (Xu et al., 2022) |
| Downward Shortwave Radiation (DSR) | 0.05° | daily | Model input | (Zhang et al., 2014b) |
| surface all-wave net radiation (Rn) | 0.05° | daily | Model input | (Jiang et al., 2015; Yin et al., 2023) |
| Evapotranspiration (ET) | 1km | 8 day | Model input | (Xie et al., 2022) |
| Fractional Vegetation Coverage (FVC) | 1km | 8 day | Model input | (Jia et al., 2015) |





| normalized difference vegetation index (NDVI) | 250m | 8 day | Model input | (Xiong et al., 2023) |
|---|---|---|---|---|
| Leaf are index (LAI) | 250m | 8 day | Variable selection | (Ma and Liang, 2022) |

### 2.2.3 FLUXCOM

The FLUXCOM initiative (www.fluxcom.org, last access: 7 March 2025) aims to improve the comprehension of the diverse sources and aspects of uncertainties in empirical upscaling, ultimately

providing an ensemble of machine learning-based global flux products to the scientific community (Jung et al., 2019). It presents two product versions: one derived exclusively from MODIS satellite observations (FLUXCOM_RS) and another that integrates meteorological data from global climate forcing datasets (FLUXCOM_RS + METEO). The spatial resolution and temporal coverage of FLUXCOM_RS are constrained by its dependency on MODIS data. Moreover, both products omit unvegetated regions,

including barren landscapes, permanent snow or ice, water bodies (such as Antarctica and Greenland), vast deserts (notably the Sahara), and much of the Tibetan Plateau. In this study, FLUXCOM_RS was chosen for its comparatively higher spatial resolution and marginally superior accuracy over FLUXCOM_RS + METEO. This version was utilized to assess the estimated H values derived from the LSTM model, with the annual data being interpolated to a 1 km spatial resolution for consistency.

**2.3 Reanalysis data**

In this study, we utilized three reanalysis datasets: ERA5, ERA5-Land, and MERRA2. The ERA5-Land was employed to evaluate the performance of the generated daily Tsa, while all three reanalysis products were used to assess the accuracy of the generated daily H product. For the period from 2000 to 2020, all product values were initially resampled to a 1 km resolution using the bilinear interpolation method

(downward fluxes considered positive). Below are detailed descriptions of these datasets:

(1)  ERA5

ERA5 (https://www.ecmwf.int/en/forecasts/dataset/, last access: 7 March 2025) represents the latest iteration of the ERA reanalysis series (Hersbach et al., 2020). With its 1-hour intervals and 31 km spatial resolution, ERA5 provides enhanced spatiotemporal precision over its predecessor, the ECMWF Interim

Re-Analysis (ERA-Interim). Its parameters have been widely validated and exhibit strong performance



across diverse applications (Li et al., 2022a; Tarek et al., 2020; Liang et al., 2022). In this study, we converted the hourly H values from ERA5 to local time and aggregated them into daily values, which were then compared with estimated H values against in-situ measurements.

(2) ERA5-Land

The ERA5-Land (https://www.ecmwf.int/en/era5-land, last access: 7 March 2025) offers a higher spatiotemporal resolution of 1 hour and 9 km. It is produced through high-resolution global numerical integrations of the ECMWF land surface model, using downscaled meteorological forcing from the ERA5 climate reanalysis. The uncertainties present in ERA5-Land are inherited from the ERA5 dataset (Muñoz-Sabater et al., 2021). In this study, we used the daily Tsa and H values from ERA5-Land for

comparison with our estimated Tsa and H values. We converted all product values to local time and computed daily averages for comparison against in-situ measurements.

(3) MERRA2

The MERRA2 (https://gmao.gsfc.nasa.gov/reanalysis/MERRA-2/, last access: 7 March 2025) is the most recent atmospheric reanalysis from NASA Global Modeling and Assimilation Office (GMAO) at

the modern satellite era. MERRA2 continues the climate record of its predecessor, MERRA, with enhancements from the updated Goddard Earth Observing System (GEOS) model and analysis program (Gelaro et al., 2017). It offers a spatial resolution of $1 / 2° \times 2 / 3°$ on an hourly basis. In this study, we aggregated the hourly H data into daily values after converting them to local time.

**3 Methods**

Figure 2 presents the flowchart of this study. Initially, four GLASS products (LST, DLW, DSR, and NDVI) along with GMTED2010 DEM data (including elevation, slope, and aspect) were used to estimate Tsa through an RF model. Subsequent analysis involved in-situ Tsa measurements and eight additional GLASS products (LAI, DSR, DLW, FVC, Rn, ABD, ET and NDVI) to identify the optimal variables for estimating H through two methods: Variance Inflation Factor (VIF) and Pearson Correlation Analysis

(referred to as Pearson). Based on the analyses, five GLASS products (DLW, Rn, FVC, ET, and ABD) and the estimated Tsa values were applied to derive H using LSTM models to account for the temporal variation of H. Considering the unavailability of ABD data during the polar night, two models were developed: mod1 for regions with ABD data and mod2 for those without, with the latter differing from

mod1 only in the exclusion of ABD. The performance of all models was subsequently evaluated against

in-situ measurements and other products, comparing the efficacy of H estimates across three different

methods: RF, Deep Belief Network (DBN), and Transformer.

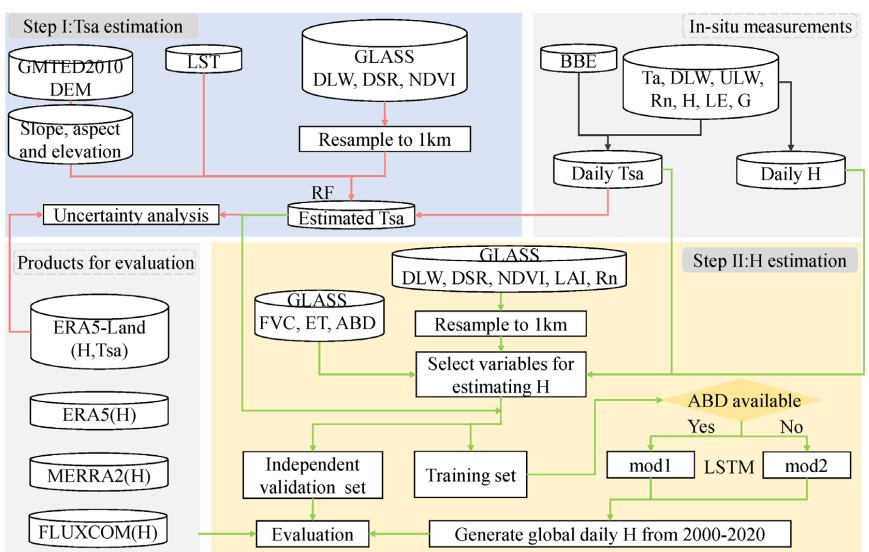

**Figure 2.** The flowchart of this study.

### 3.1 Model building of the daily Tsa

Based on the previous studies mentioned in the Introduction section and multiple experiments, the Tsa

was estimated as:

$$Tsa = f(LST, DLW, DSR, NDVI, elevation, slope, aspect, doy) \tag{3}$$

Where doy is the day of the year.

Afterwards, the Tsa estimated model was built by using the RF method (Breiman, 1996). The RF is a

widely used non-linear machine learning algorithm that constructs an ensemble of regression or

classification trees. It has gained popularity in parameter prediction and estimation due to its high

accuracy, ease of implementation, low computational cost, and fast processing speed (Babar et al., 2020;

Liang et al., 2023; Li et al., 2022a; Jiang et al., 2023). In this study, RF regression was applied, with the

final results determined by averaging the ensemble of regression output (Fig.4a). To address common

machine learning challenges such as under-fitting and over-fitting, several key hyper-parameters were

fine-tuned, including the number of trees in the forest <n-estimators>, the maximum depth of each tree

<max depth>, minimum number of samples required to split a node <min samples split>, minimum

number of samples per leaf < min samples leaf >, and etc (Babar et al., 2020). After plenty of experiments,

we identified four hyper-parameters as critical for estimating Tsa. To mitigate overfitting, we adopted a

circular approach that minimizes the root mean-squared error (RMSE) between the training and testing

phases in the RF model, in accordance with the methodology proposed by Li et al. (2022a). Consequently,

the optimal hyper-parameters for the RF model were ascertained using these two strategies, and the

results are presented in Table 4. This model is implemented on Scikit-learn toolbox (Pedregosa et al.,

2012) on the Python platform within a Microsoft Windows 10 system with 32 GB of memory.

**Table 4.** Hyper-parameter settings used to identify optimal model for estimating Tsa. The three values in
brackets for each Hyper-parameter of every model represent the start, interval, and end values,
respectively, the values in parentheses represent the value of the confirming hyper-parameter.

| n-estimators | max depth | min samples split | min samples leaf |
|---|---|---|---|
| [50,10,110] (80) | [10,5,50] (25) | [2,5,22] (7) | [2,5,22] (2) |

### 3.2 Model building of the daily H

#### 3.2.1 Land surface parameters selectin

Previous studies indicate that H is influenced by a variety of land surface parameters (Nayak et al.,

2022; Wulfmeyer et al., 2022). In this study, nine pertinent parameters were selected to determine the

optimal variables for H estimation. These included three vegetation-related parameters (LAI, NDVI, and

FVC) and six radiation-related parameters (Tsa, DLW, Rn, ABD, DSR, and ET). The significant

correlations among these parameters are well-established; for instance, DSR is frequently utilized to

calculate Rn, while both NDVI and FVC are indices associated with LAI (Xiong et al., 2023; Jiang et al.,

2015; Jiang et al., 2023). To mitigate multicollinearity and incorporate these interrelated variables, this

study employed two statistical methods for correlation analysis—VIF and Pearson—to identify the

optimal parameters for estimating H. Further details on these methods are provided below.

The VIF serve as a critical statistical metric for identifying multicollinearity within regression models,

can quantify the degree of linear correlation between one independent variable and the rest (Jiao et al.,

2017; Rehman et al., 2024), thereby enhancing the model's explanatory power and predictive accuracy.

As the VIF value rises, so does the degree of collinearity. A VIF value exceeding 10 signals significant





collinearity, warranting the variable's exclusion. The method for calculating the VIF is detailed in Eq.

325 (4):

$$VIF_i = \frac{1}{1-z_i^2} \tag{4}$$

Where $z_i^2$ is the coefficient of determination when the $i$th independent variable is regressed against

all other independent variables.

Pearson are used to measure the strength and direction of the linear relationship between two

continuous variables (Pearson, 1896). This method is widely used in various fields to identify and

quantify relationships between variables, understand data patterns and develop predictive models (Wei

et al., 2022; Yan et al., 2023). The calculation method is presented below:

$$r = \frac{\sum_{i=1}^n (M_i - \bar{M})(N_i - \bar{N})}{\sqrt{\sum_{i=1}^n (M_i - \bar{M})^2}\sqrt{\sum_{i=1}^n (N_i - \bar{N})^2}} \tag{5}$$

Where $M_i$ and $N_i$ are single sample points of variables $M$ and $N$, $\bar{M}$ and $\bar{N}$ are the mean of the

variables $M$ and $N$. The correlation coefficient $r$ ranges from −1 to 1. If $r$ is greater than 0, it means that

the two variables are linearly positive correlated, and vice versa. The absolute value of $r$, denoted as $|r|$

value, measures the strength of the linear correlation: the closer $|r|$ is to 1, the stronger the linear

correlation between the two variables. Conversely, if the $|r|$ value is closer to 0, there is little to no linear

correlation between the two variables.

Figure 3 presents the results of the multi-collinearity analysis conducted on nine land surface

parameters using the two aforementioned methods. Note that the Tsa values were obtained from in-situ

measurements which considered as "true values", while the other eight parameters were derived from the

GLASS product suite. As depicted in Fig. 3a (orange bars), the VIF values for DSR, FVC, Rn, and NDVI

surpass the threshold of 10, indicating the presence of multicollinearity. Given the functional

relationships among FVC, NDVI, and LAI, coupled with the Pearson correlation outcomes shown in

Figure 3b, FVC was chosen due to its notably negative correlation (−0.12) in comparison to LAI (0.02)

and NDVI (−0.05). Despite DSR exhibiting a higher Pearson coefficient than Rn, Rn was preferred for

its applicability in nocturnal conditions and its enhanced predictive capability, potentially owing to Rn's

reduced uncertainty relative to DSR, as evidenced in our experimental findings. Ultimately, six land

surface parameters were selected, with none exhibiting multicollinearity issues, as illustrated in Figure

3a (green bars).

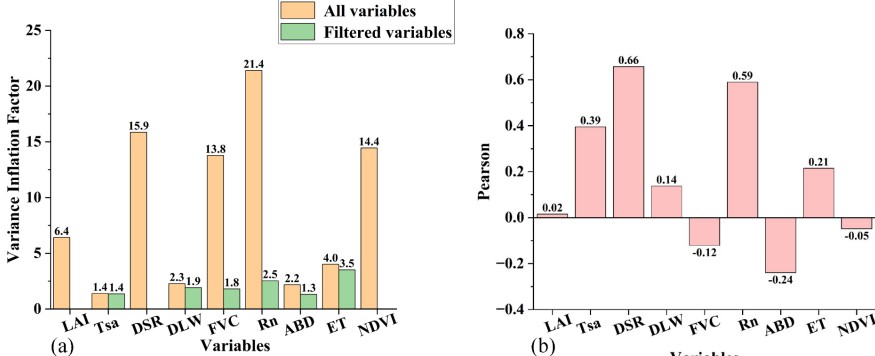

**Figure 3.** The results of the multi-collinearity analysis for nine land surface parameters using (a) the VIF
method and (b) the Pearson method. The orange bars represent pre-filter variables, while the green bars
represent post-filter variables.

### 3.2.2 Modeling building

According to the results presented in Section 3.2.1, six variables were used in estimating daily H.

Furthermore, due to the unavailability of ABD data during the polar night, two models were developed:

one for areas with ABD data (designated as mod1) and another for areas without ABD data (designated

as mod2). Thus, the H estimation model is expressed mathematically as follows:

$$H = \begin{cases} f(Tsa, ABD, DLW, FVC, R_n, ET), areas \ with \ ABD \\ f(Tsa, DLW, FVC, R_n, ET), areas \ without \ ABD \end{cases} \tag{6}$$

Subsequently, the LSTM was used to constructed the estimated model of daily H. LSTM network, an

extension of the traditional Recurrent Neural Network (RNN), is a feedforward network with a feedback

loop and internal memory (Lyu et al., 2016). LSTM addresses the issues of exploding and vanishing

gradients by incorporating memory blocks and gating mechanisms. Each layer of the network contains

three gates—input, forget, and output gates—as well as a cell state (Xiong et al., 2023). This design

allows LSTM to maintain long-term memory more effectively than RNNs, making it better suited for

handling long-term dependencies. Further details have been provided by Ma et al (2022). In this study,

the two LSTM models have same structure and hyper-parameters, each one was designed with one input

layer, two LSTM layers consisting of 400 and 250 neurons, and one regression layer. The structure is

illustrated in Fig.4(b). In model training stage, after extensive experimentation, the Adam optimizer was

selected and the parameters of batch size, max epochs and learning rate were set to 16, 100 and 0.001,

respectively. The entire process was implemented in Python platform using the LSTM module from the



Keras toolbox (https://github.com/keras-team/keras/), on a Microsoft Windows 10 system with 32 GB of

memory.

## 3.3 Comparing daily H estimated model

Three methods were picked for their representativeness to compare with LSTM in estimating daily H,

including RF, DBN and Transformer. The introduction to RF is provided in Section 3.2 and its optimal

hyper-parameters setting is provided in Table.5. The details of the DBN and Transformer methods are

described below, with their structures illustrated in Fig. 4 (c-d).

**Table 5.** Same as Table3 but for estimating H.

| | Hyper-parameters | | | |
|---|---|---|---|---|
| | n-estimators | max depth | min samples split | min samples leaf |
| mod1 | [50,10,110] (10) | [10,5,50] (10) | [2,5,22] (17) | [2,5,22] (2) |
| mod2 | [10,10,100] (80) | [10,5,30] (5) | [2,5,22] (5) | [2,5,22] (5) |

(1) Deep Belief Network

The DBN, introduced by Hinton et al. (2006), has become one of the widely used deep learning models

for estimating lad surface parameters (Zang et al., 2019; Li et al., 2017; Shen et al., 2020). As a Bayesian

probabilistic generation model, DBN typically consists of multiple Restricted Boltzmann Machines

(RBMs) and a backpropagation (BP) layer as Fig.4 (a) shows. The RBM, an energy-based model,

addresses challenges like local optima and gradient vanishing by pre-training neural networks (Hinton et

al., 2006). This is done by adjusting model parameters using contrastive divergence, ensuring the

probability distribution of visible units closely aligns with the input data (Shen et al., 2020). An RBM

generally consists of a visible layer and a hidden layer, with the hidden layer of one RBM acting as the

visible layer for the subsequent RBM in the DBN (Shen et al., 2018). The BP layer is typically employed

for classification or regression tasks. In this study, the batch size, activation function, network structure,

and learning rate were optimized. Following extensive experimentation, the DBN model was constructed

with one RBM layer and one hidden layer, and the optimal hyper-parameter combination is provided in

Table 6.

Table 6. The hyper-parameter setting for two models. The optimal parameter values for mod1 and mod2
are listed in the last two columns

| Hyper-parameters | values | The optimal parameters |
|---|---|---|





|  |  | mod1 | mod2 |
|---|---|---|---|
| batch size | 16,32,64,128,256,512,1024,2048 | 32 | 16 |
| activation function | Relu, Tanh, Sigmoid | Relu | Relu |
| neurons of hidden layer | 16,32,64,128,256,512,1024,2048 | 256 | 128 |
| Learning rate | 0.01,0.05,0.001,0.005 | 0.05 | 0.05 |
| Learning rate of RBM | 0.01,0.05,0.001,0.005 | 0.05 | 0.005 |

(2) Transformer

The Transformer, introduced by Vaswani et al. (2017), is a sequence-to-sequence model based on a

self-attention mechanism. This design enables it to focus on different parts of the input sequence simultaneously, and leads to better performance with fewer parameters in certain cases. Therefore, the Transformer excels at parallel processing of long sequences, which leads to significant improvements in training efficiency, easily capturing long-term dependencies while being less vulnerable to gradient-related issues (Tay et al., 2020). Currently, The Transformer model and its variants have been widely

used in time series forecasting with good results (Lim et al., 2021; Zhou et al., 2021). Typically, the Transformer is composed of an encoder-decoder structure, where both the encoder and decoder utilize layers of self-attention and feedforward networks. In this study, the encoder layers applied two transformer blocks, each containing a multi-head self-attention mechanism with 70 heads and a head size of 10 along with a fully connected feedforward network with 32 neurons. However, the decoder layer

primarily utilized a Multilayer Perceptron (MLP), which was more suitable for the task. The construction of the Transformer network illustrated in Fig. 4 (b).

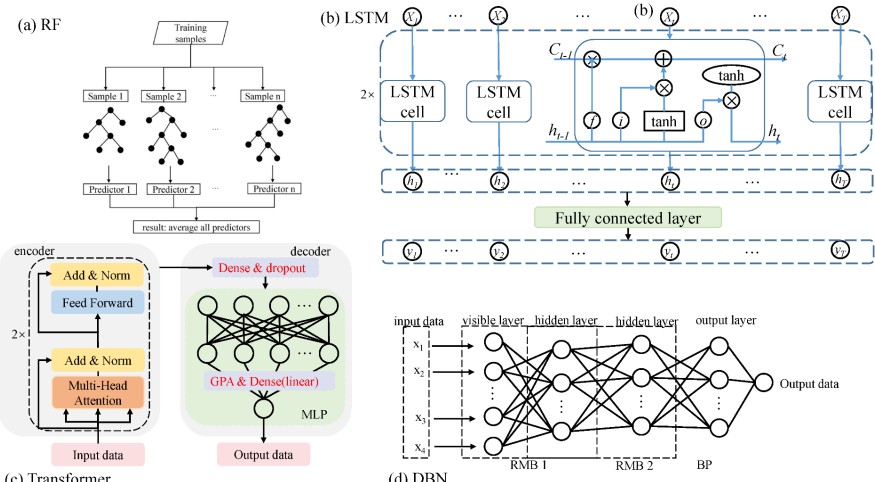

**Figure 4.** The structure diagram of (a) RF model, (b)LSTM, (c) Transformer model and (d) DBM model in this study. Add and Norm are the Residual Connection and Layer Normalization. GPA represents the global average-pooling operations.

## 3.4 Evaluation approaches

Three statistical measures were used to represent the validation accuracy: RMSE, Mean Absolute Error (MAE), and the **coefficient of determination** ($R^2$).

$$MAE = \frac{\sum_{i=1}^{N} |(Y_i - X_i)|}{N} \tag{7}$$

$$RMSE = \sqrt{\frac{1}{N} \sum_{i=1}^{N} (X_i - Y_i)^2} \tag{8}$$

$$R^2 = 1 - \frac{\sum_{i=1}^{N} (X_i - Y_i)^2}{\sum_{i=1}^{N} (X_i - \bar{X})^2} \tag{9}$$

where $Y_i$ and $X_i$ are the estimation and the measurement values of the $i_{th}$ group of samples, and $N$ represents the number of samples.

## 4 Results

### 4.1 Uncertainty quantification of Tsa model

As described in Section 2.1, the measurement values from 2000 to 2017 were used for training, while data from the subsequent two years, 2018 and 2019, served as independent validation samples to assess the model's performance. Figure 5 illustrates the training and independent validation accuracy of the



estimated Tsa RF model. The overall accuracy of the estimated Tsa on independent validation samples is

satisfactory, with an RMSE of 1.459 K, a MAE of 1.071 K, and an $R^2$ of 0.53 (Fig. 5b). And there is a

slight deviation from the training accuracy, as indicated by an increase of 0.757 K in the RMSE (Fig. 5a).

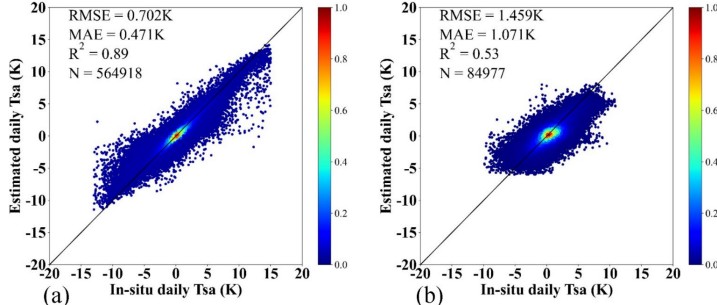

**Figure 5.** The overall results of the (a) training and (b) independent accuracy of validation by using RF method during 2018-2019.

Subsequently, the accuracy of the estimated Tsa using the RF model was compared to that of GLASS

and ERA5-Land, both of which possess relatively high spatial resolution, using the same set of

independent validation samples (n = 83,284). The Tsa values for these two products were calculated by

subtracting Ta from LST. The estimated Tsa (Fig. 6a) achieved an RMSE of 1.46 K, a MAE of 1.073 K,

and an $R^2$ of 0.52, significantly outperforming GLASS and ERA5-Land (Fig. 6b and c), which yielded

RMSEs of 2.238 K and 2.037 K, MAEs of 1.667 K and 1.394 K, and $R^2$ values of 0.11 and 0.32,

respectively. Moreover, the estimated Tsa values more closely align with the 1:1 line in the scatter plot,

whereas GLASS Tsa displays a divergent pattern and ERA5-Land significantly underestimates values

below 0 K. Such discrepancies likely stem from variations in land cover types, highlighting the robust

performance of the proposed RF model in estimating Tsa.

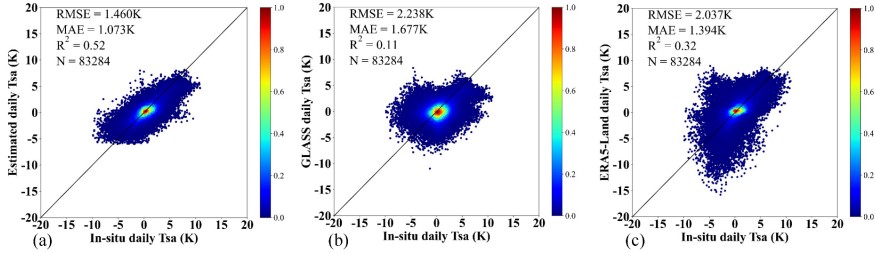

**Figure 6.** Comparison of the validation accuracy against in-situ measurements by using common samples in Tsa from (a) the estimated values by using RF model in this study, (b) GLASS and (c) ERA5-Land.



For further exploration, the validation accuracies of the estimated Tsa model and two other products at a daily scale were further examined across various conditions, including five elevation ranges (0–200m, 200–500m, 500–1000m, 1000–1500m and >1500m), five NDVI ranges (0–0.2, 0.2–0.4, 0.4–0.6,0.6–0.8 and 0.8–1), six slope ranges (0-2°, 2-4°, 4-6°, 6-8°, 8-10° and >10°) and nine land cover types (BSV, CRO, CVM, FOR, GRA, IAS, SAV, SHR and WET). The evaluation results are presented in Fig.7.

Overall, the RF estimated model exhibited superior accuracy in various conditions, followed by ERA5-Land, aligning with findings from Fig. 6. In terms of terrain factors like elevation and slope (Fig. 7 a and g), the RF model showed significant improvements, especially within the 500–1000 m and >1500 m elevation ranges, and the 2–6° and >8° slope ranges, with RMSE improvements of approximately 1 K for elevation and between 0.7 K and 1.4 K for slope. Regarding NDVI (Fig.7 e), a notable improvement was observed in the 0–0.2 range, with an RMSE of ~1 K. These outcomes suggest that the RF model more accurately reflects Tsa variations influenced by vegetation and terrain, which are key factors mentioned in the Introduction. Additionally, the performance across different land cover types was evaluated in Fig. 7 c. The RF model performed well across all types, except for a site in CVM, which had an RMSE of approximately 2.2 K and an MAE of 1.8 K, slightly higher than GLASS and ERA5-Land by 0.3 K and 0.5 K in RMSE, respectively. Among the eight land cover types evaluated, the RF model demonstrated exceptional performance, especially in areas with high albedo (IAS) and those subject to seasonal variations (WET). In these cases, ERA5-Land recorded an RMSE of approximately 3.3 K for IAS, while GLASS reported around 2.9 K, implying the need for caution when applying these products to studies of icy regions. Regarding the BSV, the accuracy of the RF model was on par with ERA5-Land and surpassed that of GLASS, suggesting that the RF model and ERA5-Land effectively incorporate critical information.

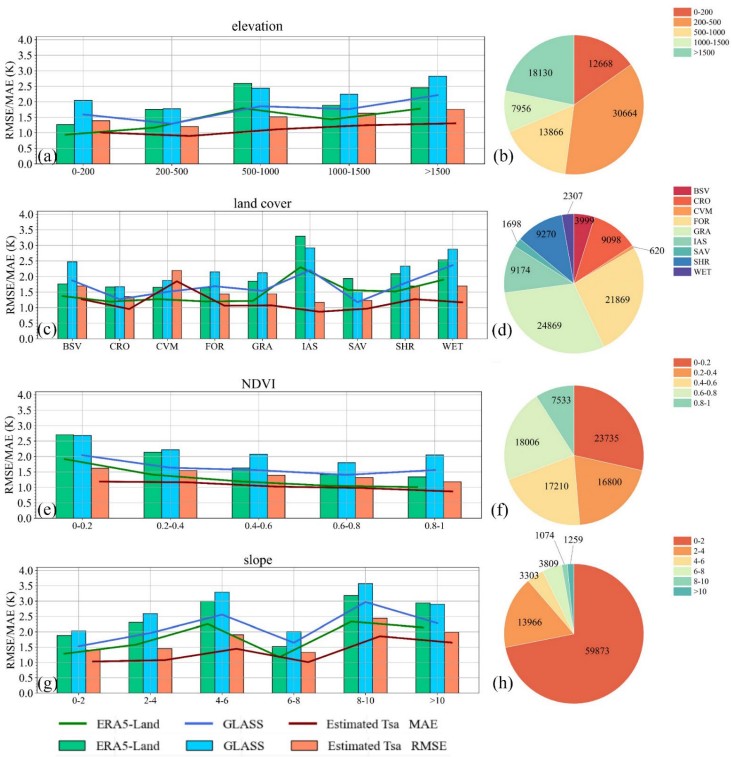

470

**Figure 7.** The validation accuracy of estimated Tsa, ERA5-Land and GLASS across different conditions: (a) elevation within 0–200m, 200–500m, 500–1000m, 1000–1500m and >1500m, (b) land cover described in Fig.1, (c) NDVI with an interval of 0.2 and (d) slope with an interval of 2°. The pie charts in (b), (d), (f) and (h) display the corresponding sample sizes for each condition.

475    In summary, the accuracy of the RF model and the two comparison products varied significantly under different conditions, with the RF model consistently outperforming the others as indicated by the lowest RMSE and MAE in almost all cases. Consequently, the RF model was used to generate daily Tsa values globally from 2000 to 2020 for the calculation of daily H.

### 4.2 Evaluation of model accuracy for H estimation

480    Table 7 presents the training and validation accuracy of estimated daily H from two LSTM models against in-situ measurements using the same samples. The independent validation results of two models were different but acceptable, with RMSEs of 25.533 and 27.051 Wm$^{-2}$, MAEs of 18.641 and 20.034 Wm$^{-2}$, and R$^2$ of 0.54 and 0.51. Compared to the training accuracy, the validation results showed a slight increase in RMSE (3.24 and 2.232 Wm$^{-2}$) and MAE (2.147 and 1.469 Wm$^{-2}$), but these differences were



considered acceptable. Additionally, incorporating ABD into the model improved accuracy, reducing RMSE to 1.518 Wm$^{-2}$ and MAE to 1.393 Wm$^{-2}$, underscoring the significance of ABD. Overall, all models exhibited satisfactory performance, as evidenced by their comparable training and validation results.

**Table 7.** The training and independent validation accuracy of two LSTM models based on the same samples, both with and without the use of ABD data. Units of RMSE and Bias are Wm$^{-2}$.

| | Training (No. of samples = 121,542) | | | Independent validation (No. of samples = 97,982) | | |
|---|---|---|---|---|---|---|
| | RMSE | MAE | $R^2$ | RMSE | MAE | $R^2$ |
| mod1 | 22.293 | 16.494 | 0.71 | 25.533 | 18.641 | 0.54 |
| mod2 | 24.819 | 18.565 | 0.64 | 27.051 | 20.034 | 0.51 |

Due to the lack of ABD data during the polar night, two models were developed. In the final validation phase, the polar night results from model 1 were substituted with those from model 2, as depicted in Fig.8. The overall validation accuracy was deemed satisfactory, with an RMSE of 25.54 Wm$^{-2}$, MAE of 18.649 Wm$^{-2}$ and $R^2$ of 0.54. Furthermore, the spatial distribution of the performance of daily H, represented by the RMSE at each validation site, was calculated and illustrated in Fig 9. It was observed that the LSTM model exhibited the highest level of robustness on a global scale, with 80% of the sites (107 sites) reporting an RMSE below 30Wm$^{-2}$ (indicated in red and orange in Fig. 9). Nonetheless, some sites (eight sites) displayed suboptimal performance with RMSE values exceeding 40 Wm$^{-2}$.

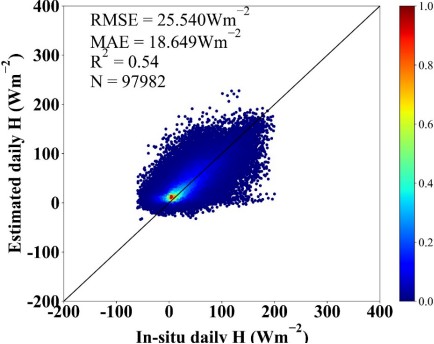

**Figure 8.** The overall validation accuracy of the estimated H based on all independent in-situ validation samples. The values were obtained by replacing the results for areas with missing ABD in mod1 with those from mod2.

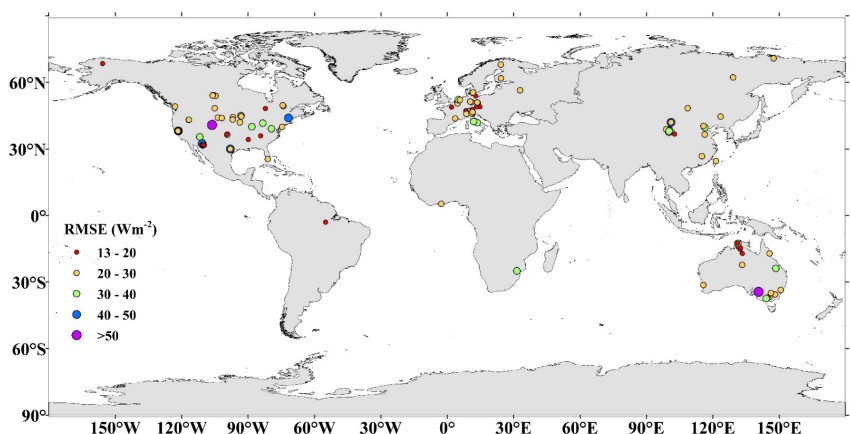

**Figure 9.** The spatial distribution of the validation accuracy of all sites (represented by RMSE).

Afterwards, three different methods were employed to estimate H for a comparative analysis with the accuracy of the LSTM models. The independent validation results were presented in Fig. 10 based on the same samples as Fig.8. The three methods produced closely aligned results, with RMSE values ranging from 25.341 to 26.01 Wm$^{-2}$, MAE values between 18.757 and 19.165 Wm$^{-2}$, and R$^2$ values from 0.52 to 0.55, compared to the LSTM model's RMSE of 25.54 Wm$^{-2}$, MAE of 18.649 Wm$^{-2}$ and R$^2$ of 0.54 (as shown in Fig.8). However, all models exhibited varying degrees of underestimation for high values and overestimation for low values. While this issue was particularly pronounced in the Transformer and RF methods, the DBN and LSTM models demonstrated relatively better performance, albeit with similar tendencies. Remarkably, the LSTM model surpassed the DBN model, achieving improvements of 0.47 Wm$^{-2}$ in RMSE and 0.516 Wm$^{-2}$ in MAE. To further clarify the performance of these models, we examined each site and randomly selected three sites to illustrate the temporal variations in the values of H based on these four methods and in-situ measurements, compared against validation samples. As shown in Fig.11, the LSTM model effectively captures the temporal variation of H in relation to in-situ measurements, while the other three models exhibit relatively poorer performance on certain days. A notable mismatch is observed in the RF, DBN, and Transformer models around the 268th day of 2012 at Lath_AU-Dry (Fig.11 a), with RF displaying only a single value and the other two methods showing underestimation on those days. Furthermore, RF and DBN exhibit opposing trends comparing to in-situ measurements during the 152nd and 218th days of 2011 at Lath_US-SRC (Fig.11 c). The variations of the four models at Lath_US-Dia (Fig.11 b) generally coincide with in-situ measurements,

but RF shows slight overestimation around the 248th day of 2011 and DBN underestimates before the

196th day of 2012. Therefore, the LSTM model demonstrated superior performance, effectively

mitigating the challenges of overestimating low values and underestimating high values in this study,

likely due to its incorporation of time series information.

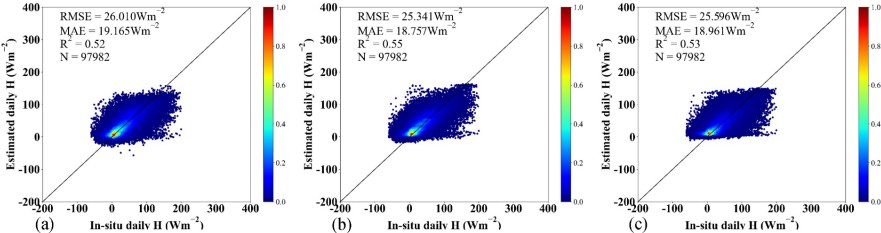

**Figure 10.** Validation accuracy against in-situ measurements using the common validation samples as

LSTM for (a) DBN, (b) RF and (c) Transformer methods.

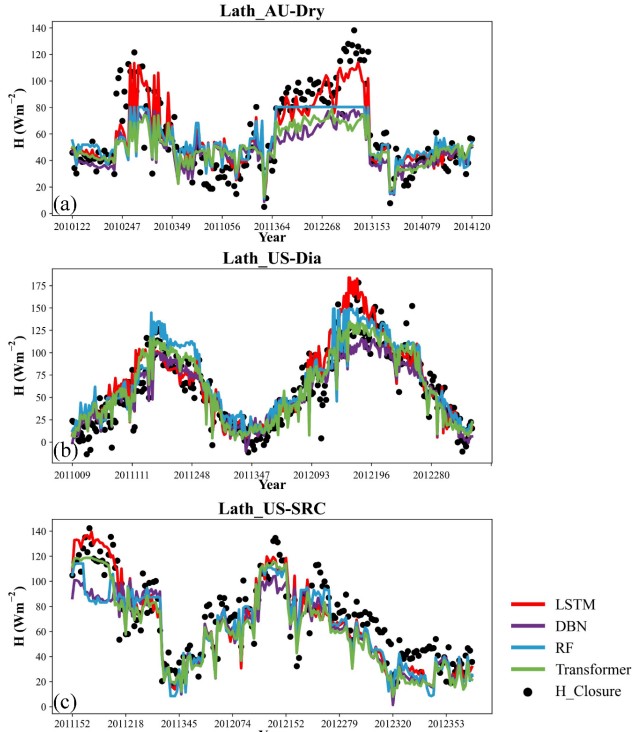

**Figure 11.** Temporal variations in the values of H based on the LSTM (red line), DBN (purple line), RF
(blue line), Transformer (green line) and in-situ measurements (black dot) using the validation samples
at (a) Lath_AU-Dry, (b) Lath_US-Dia and (c) Lath_US-SRC. Note that the time given on the abscissa is

not continuous in (a)一(c)

In summary, the LSTM models employed for estimating daily H, which integrate the estimated Tsa

and other GLASS products, have shown satisfactory accuracy. Consequently, this method is deemed appropriate for global 1km resolution mapping of daily H, establishing it as a viable and dependable approach for these applications.

**4.3 Daily H product generation**

In this study, daily H estimates were generated globally for the period 2000–2020 by integrating two LSTM models. Specifically, mod2 was applied under polar night conditions when ABD data were unavailable. To assess the accuracy of the estimated H, we examined its spatial and temporal variations and compared the daily estimated H values with other existing products, as outlined below.

**4.3.1 The spatial and temporal variation of H**

Figure 12 presents the results of calculating monthly average values across different latitude zones with a 10° range for all years. It reveals that the variation in H demonstrates distinct seasonal patterns, with higher values observed during the summer months in both hemispheres. This trend aligns with Tsa variations, highlighting the impact of solar radiation on surface properties, which in turn affects the 550 energy balance and flux dynamics (Jiang et al., 2022). Specifically, high H values are found in three regions: between 30–60°N from May to August, 20–50°S from January to March, and 10–50°S from October to December, peaking in January (82.15 Wm$^{-2}$) at 40–50°S. Conversely, winter months at higher latitudes exhibit low values, with the lowest recorded in June (–3.8 Wm$^{-2}$) at 60–70°S. Generally, H values in polar regions remain below 15 Wm$^{-2}$, occasionally dropping below 0 Wm$^{-2}$. Nevertheless, in 555 March, April, and September, H values surpass 40 Wm$^{-2}$ around 80°S. The scarcity of observation sites in polar regions might increase uncertainty in our models, particularly in the South Pole region (>70°S), thus caution is advised when using H values.

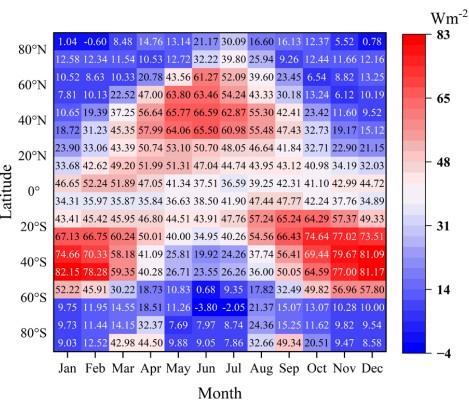

**Figure 12.** Variation of monthly H in latitude zones (10°) and months zones from 2000–2020.

To better illustrate the spatial and temporal variations in global H during 2000–2020, Fig. 13 displays

the anomaly values of land surface H across latitude zones (1°) for each day. There is a clear annual

pattern influenced by the sun's position evident across these years. The position of the sun directly

influences the distribution of DSR, which in turn affects Tsa, and ultimately altering the distribution of

H. Additionally, a distinct cyclic trend is noticeable in both latitudinal and temporal variations, reflecting

seasonal changes across latitudinal zones and underscoring dynamic shifts in H distribution over time.

These shifts may result from a combination of regional climatic changes, land surface properties, and

interactions with atmospheric processes. These findings underscore the importance of long-term satellite-

based remote sensing for capturing spatiotemporal variations in land-atmosphere energy exchanges. Such

observations are essential for understanding the mechanisms behind energy flux dynamics and their

sensitivity to environmental and climatic changes.

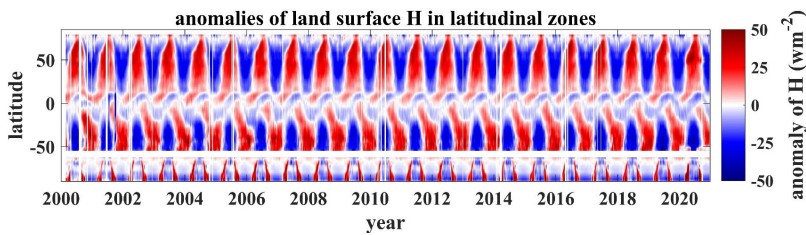

**Figure 13.** The anomalies of land surface H in latitude zones (1°) at daily scales from 2000–2020.

    In summary, the spatial and temporal variations observed in the estimated H data align with theoretical

expectations, yet they necessitate further validation. To this end, we conducted a comparison of the

estimated H with other existing products to provide a more thorough evaluation.





### 4.3.2 Inter-comparison with other products

Three reanalysis products (MERRA2, ERA5 and ERA5-Land) and one remotely sensed-based product (FLUXCOM) were further compared. Fig. 14 (a1-a5) illustrates the spatial distributions of these four products and the estimated H on the 121st day of 2010 at a global scale. The spatial distribution of the estimated H is logical and closely resembles that of MERRA2, ERA5-Land, and ERA5, while FLUXCOM exhibits relatively lower values compared to the other products. Additionally, we provide a further comparison of the estimated H values with other products in the Tibetan Plateau region, characterized by its complex terrain, as shown in Fig. 14 (b1-b5), corresponding to the black box in Fig. 14 (a1-a5). The spatial distribution of the three reanalysis products is noticeably smoother than that of the estimated H, and FLUXCOM lacks most data in this region. The estimated H effectively captures the intricate details of the rugged terrain, thanks to its higher spatial resolution, a detail that is not as prominently reflected in the other four products (Fig.14 c1-c5). This comparison highlights the importance of high-resolution H products for accurately depicting complex landscapes.

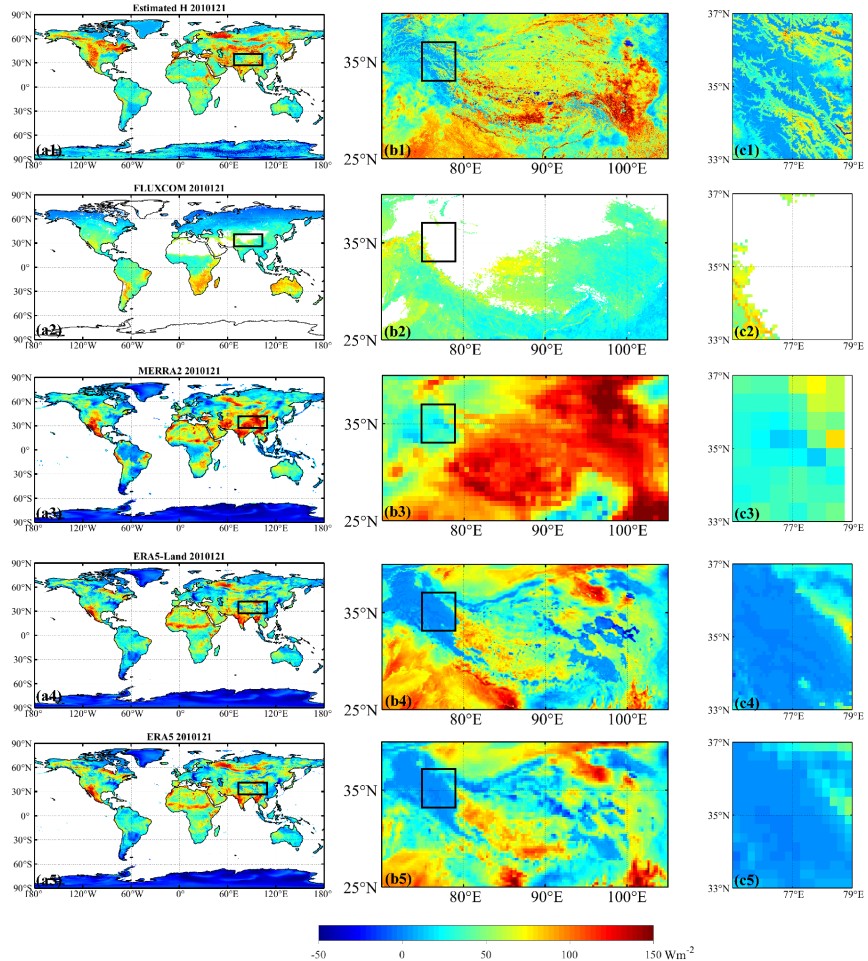

**Figure 14.** (a1−a5) display the daily values on the 121th day of 2010 for the estimated H, FLUXCOM, MERRA2, ERA5-Land and ERA5. The black box in (a1−a5) represents the location of (b1-b5) and (c1-c5) is the location of black box in (b1-b5).

Figure 14 reveals significant discrepancies in the estimated H values in certain areas when compared to the other products. Therefore, we further employ in-situ measurements to evaluate the accuracy of the estimated H values in the subsequent sections. To ensure spatial consistency, all products were interpolated to a resolution of 1 km. FLUXCOM_RS was evaluated separately as it is the sole publicly available global remote sensing product that offers an 8-day temporal resolution spanning from 2001 to 2015. In contrast, the reanalysis products feature higher temporal resolutions (hourly) and encompass a broader timeframe (1950 to the present).

**4.3.2.1 Reanalysis products**

Figure 15 illustrates the performance of the estimated H values in comparison to three reanalysis products (MERRA2, ERA5 and ERA5-Land), utilizing 97,045 independent validation samples. The independent validation results show that the estimated H values outperformed those of the three reanalysis products, achieving the lowest RMSE of 26.587 $Wm^{-2}$ and MAE of 19.191 $Wm^{-2}$. Remarkably, the estimated H exhibited significantly lower uncertainty compared to the other products, with reductions in RMSE of 9.351, 5.497 and 4.573 $Wm^{-2}$ and in MAE of 6.996, 4.342 and 3.562 $Wm^{-2}$ for MERRA2, ERA5-Land and ERA5, respectively. Moreover, the estimated H demonstrated enhanced accuracy for values approaching zero, in contrast to the significant uncertainty observed in the reanalysis products for small H values, potentially indicative of winter conditions (highlighted by the red circles in the Fig. 15 b, c and d). These findings suggest that caution is advised when employing MERRA2, ERA5, and ERA5-Land for small absolute H values.

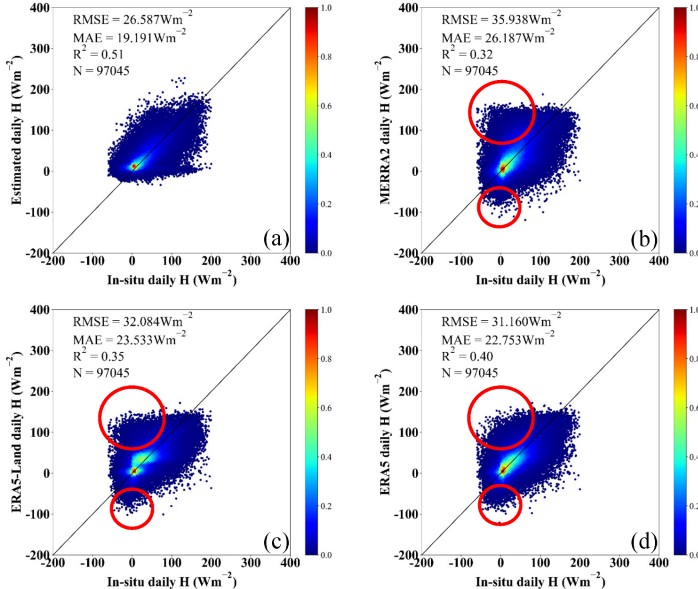

**Figure 15.** Comparison of the validation accuracy against in-situ measurements by using common samples in H from (a) the estimated H in this study, (b) MERRA2 and (c) ERA5-Land and (d) ERA5.

Additionally, to provide a more comprehensive evaluation, we compared the performance of the estimated daily H using validation samples against three other products across seven land cover types. The comparison results, depicted in Fig. 16(a–c), include RMSEs, MAEs, and $R^2$ values, while Fig. 16(d)

provides the sample sizes for each land cover category. These results demonstrate that the accuracy of the estimated H varies by land cover type, with RMSEs ranging from 23.87 to 32.39 Wm$^{-2}$, MAEs from

17.66 to 23.21 Wm$^{-2}$, and R$^2$ from 0.31 to 0.58. Overall, the estimated daily H outperformed the three other datasets, followed by ERA5 (with RMSEs between 19.05 and 42.12 Wm$^{-2}$ and MAEs between 14.61 and 31.58 Wm$^{-2}$), ERA5-Land (with RMSEs between 20.19 and 48.46 Wm$^{-2}$ and MAEs between 15.72 and 36.34 Wm$^{-2}$) and MERRA2 (with RMSEs between 28.11 and 54.42 Wm$^{-2}$ and MAEs between 21.47 and 39.46 Wm$^{-2}$). Specifically, the estimated daily H exhibited superior performance for land cover

types such as WET (27.38 Wm$^{-2}$ in RMSE and 19.71 Wm$^{-2}$ in MAE), SHR (25.29 Wm$^{-2}$ in RMSE and 18.72 Wm$^{-2}$ in MAE), GRA (23.87 Wm$^{-2}$ in RMSE and 17.66 Wm$^{-2}$ in MAE), FOR (27.86 Wm$^{-2}$ in RMSE and 19.78 Wm$^{-2}$ in MAE), and CRO (26.37 Wm$^{-2}$ in RMSE and 19.51 Wm$^{-2}$ in MAE), with the RMSE and MAE values significantly lower than those of ERA5, ERA5-Land and MERRA. However, the estimated daily H showed marginally lower performance for SAV and BSV, with all datasets yielding

relatively similar RMSEs (ranging from 25.29 to 29.7 Wm$^{-2}$) and MAEs (from 18.72 to 22.15 Wm$^{-2}$). This indicates that the estimation methods produce comparable results for these specific land cover types.

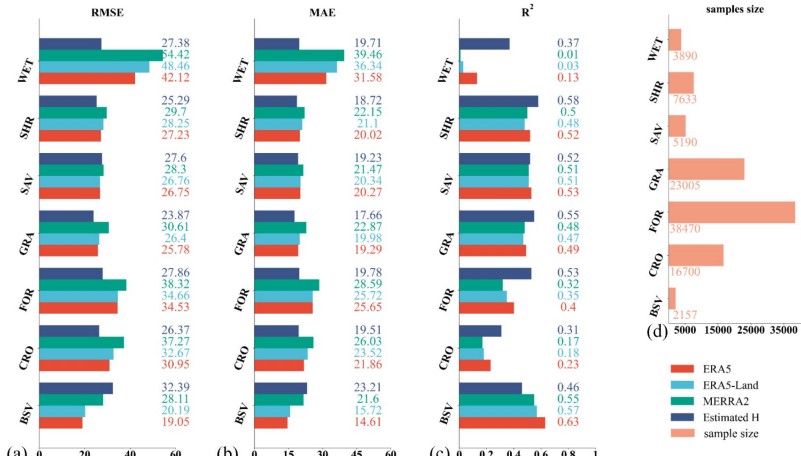

**Figure 16.** The (a) RMSE, (b) Bias, (c)R$^2$ of LSTM model with ERA5, ERA5-Land and MERRA2 in various land cover types. The corresponding sample size in different land cover provided in (d).

To further assess the performance across different regions, we compared the daily H estimates with other datasets using in-situ measurements across six continents, as illustrated in Fig. 17 (a1-a6). The comparison reveals that the estimated H achieved commendable performance in North America, Europe, Asia, and Australia, with RMSEs of 26.55, 27.15, 25.87, and 26.63 Wm$^{-2}$, respectively. Notably, the



RMSEs associated with the estimated H decreased significantly comparing with other three products,

ranging from 5.14 to 10.21 Wm$^{-2}$ in North America, 4.36 to 6.02 Wm$^{-2}$ in Europe, 7.04 to 21.5 Wm$^{-2}$ in

Asia, and 2.42 to 8.48 Wm$^{-2}$ in Australia. Conversely, the estimated H exhibited weaker performance in

South America and Africa, where the validation was constrained to a limited number of sites—

specifically, one site (N = 548) in South America and two sites (N = 1048) in Africa, as shown in Fig. 9

and Fig. 17(a7). In South America, the estimated H reported an RMSE of 19.12 Wm$^{-2}$, with ERA5

outperforming other datasets by achieving the lowest RMSE of 12.09 Wm$^{-2}$. In Africa, the difference in

RMSE between the estimated H and MERRA2 was minimal, at merely 3.03 Wm$^{-2}$, whereas the greatest

discrepancy was noted with ERA5-Land, which exhibited a difference of 6.53 Wm$^{-2}$.

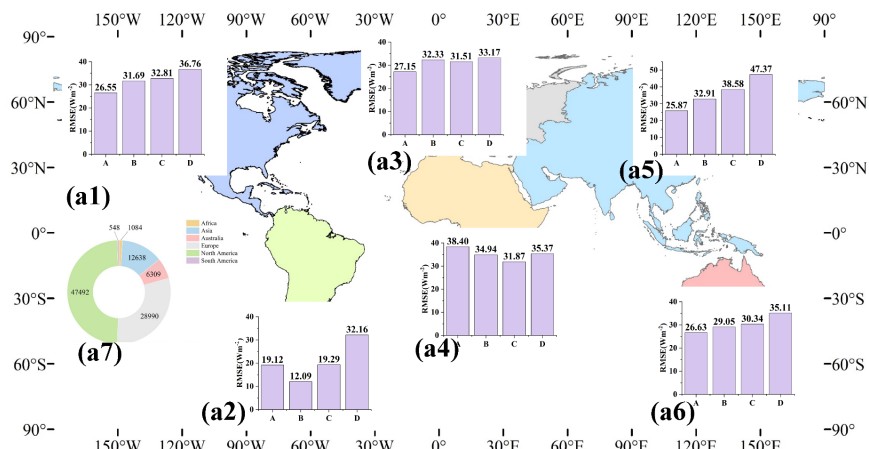

**Figure 17.** The RMSE values for four datasets across six continents: (a1) North America, (a2) South

America, (a3) Europe, (a4) Africa, (a5) Asia, and (a6) Australia. A-D represent the estimated H, ERA5,

ERA5-Land, and MERRA2, respectively. (a7) shows the corresponding sample sizes for each continent.

### 4.3.2.2 FLUXCOM

The H estimates derived from LSTM were compared with the sole publicly remotely sensed-based

product, FLUXCOM, through independent validation samples spanning 2001 to 2015 (as shown in

Fig.18). The accuracy of the H estimates surpassed that of FLUXCOM, as evidenced by lower RMSE

and MAE values of 24.5 and 18.14 Wm$^{-2}$, respectively, in comparison to FLUXCOM's RMSE and MAE

of 29.21 and 21.82 Wm$^{-2}$ (Figure 18, panels a1 and b1). Furthermore, the majority of sites with estimated

H exhibited RMSE values below 30 Wm$^{-2}$, predominantly located in Eastern Asia, European, Eastern

American, and the Northern and Southeastern regions of Australia, as depicted in Fig. 18 a2 and b2. In

contrast, the spatial distribution of FLUXCOM demonstrated significant variability across continents, with RMSE values ranging from 10 to approximately 40 $Wm^{-2}$ even within the same continent or adjacent regions.

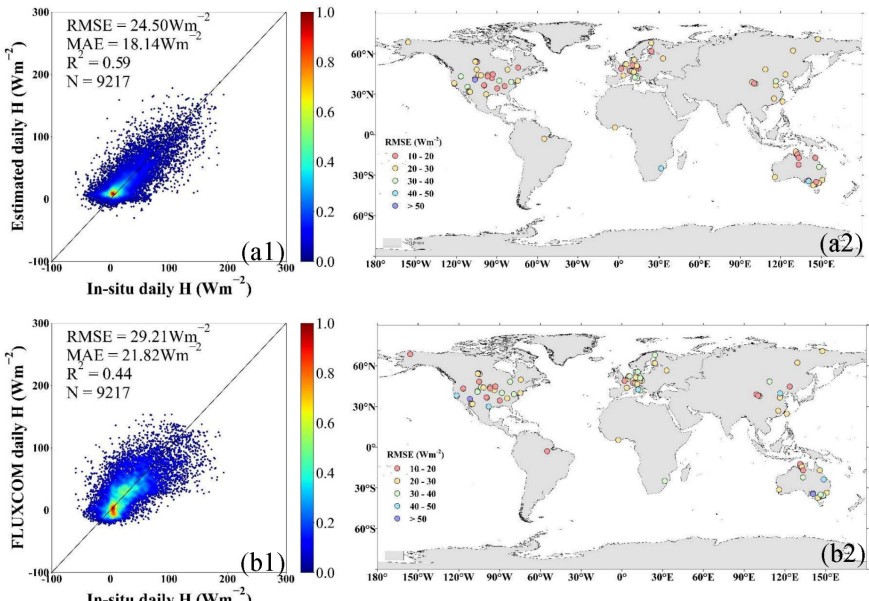

**Figure 18.** The overall validation accuracy against in-situ measurements by using common samples from
(a1) the estimated H in this study and (b1) FLUXCOM_RS. (a2) and (b2) show their corresponding spatial distribution of site overall validation accuracy (represented by RMSE).

     Based on the preceding results, the estimated H exhibits superior performance compared to FLUXCOM. To ensure a more thorough evaluation, we further assessed the validation accuracy of both products across various months, land-cover types, and elevation ranges, utilizing the same samples
depicted in Fig. 18. Here, we present the RMSE values, which have been determined to accurately reflect the performance in each scenario following an extensive evaluation. Overall, the accuracy of both products exhibited variability under different conditions, yet the estimated H consistently surpassed FLUXCOM in all scenarios. Figure 19a reveals a distinct seasonal variation in accuracy, characterized by reduced RMSE values in the winter months and increased values during the summer. A similar trend
was observed for Rn, which informed the derivation of H in this study (Yin et al., 2023). This seasonal fluctuation is likely due to seasonal differences in cloud cover and water vapor content, which influence radiation estimates and thus affect the H estimates. The disparity in RMSE values between the two products ranged from 1.86 to 7.5 $Wm^{-2}$, with the most significant differences noted in May and June. For



different land-cover types (Fig. 19 b), the estimated H demonstrated stable performance, with RMSE

values ranging from 24 to 30 $Wm^{-2}$, indicating that the LSTM method effectively captured the features

of each land cover type. In terms of accuracy for CRO and GRA, both products were comparable, with

a nominal RMSE disparity of approximately 1.5 $Wm^{-2}$. However, both products demonstrated relatively

weaker performance in SHR, with RMSEs of 29.78 $Wm^{-2}$ for the estimated H and 34.94 $Wm^{-2}$ for

FLUXCOM. Notably, the estimated H achieved significant improvements in WET and FOR, with RMSE

improvements of 7.67 $Wm^{-2}$ and 6 $Wm^{-2}$, respectively. The comparison of accuracy across five elevation

ranges is depicted in Fig. 19c. With increasing elevation, the accuracy of both products diminished. In

regions exceeding 1500 m in elevation, the RMSE values reached 30.38 $Wm^{-2}$ for the estimated H and

35.09 $Wm^{-2}$ for FLUXCOM. Conversely, at elevations below 1500 m, the estimated H maintained a more

consistent performance, with RMSE values spanning from 23.11 to 25.25 $Wm^{-2}$, in contrast to the RMSE

values of FLUXCOM, which varied from 26.17 to 32.05 $Wm^{-2}$.

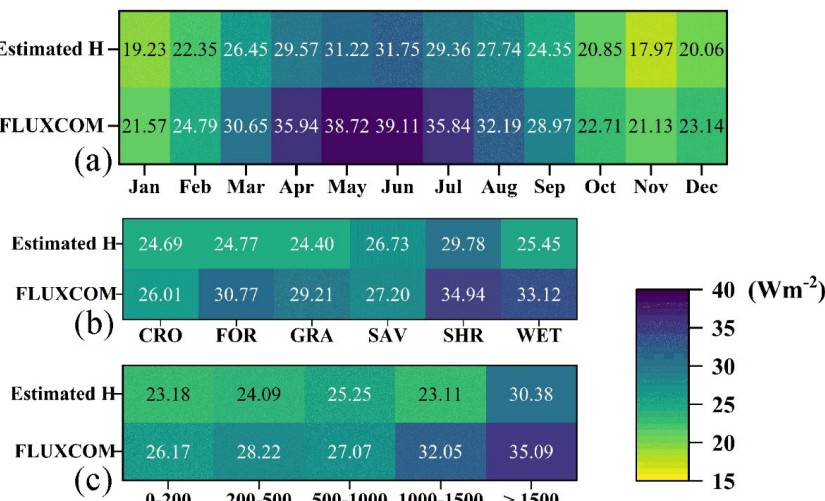

**Figure 19.** Comparison of the validation accuracy (represented by RMSE) in H under three conditions:
(a) twelve months, (b) land-cover types and (c) elevation ranges (200m, 200–500m, 500–1000, 1000–
1500m, and >1500m)

Overall, the daily H estimates over a 1 km resolution from 2000 to 2020, derived through the

application of LSTM models based on calculated Tsa, exhibit significant potential for broad application.

This potential arises from their commendable accuracy and their proficiency in capturing surface

characteristics, as compared to other existing products.





**5 Discussion**

Global H products encounter limitations, including coarse spatial resolution and significant uncertainties. Given that Tsa is a crucial factor in deriving H, this study employs it to obtain H. Nevertheless, the accuracy of Tsa calculations frequently suffers when derived from existing datasets by subtracting $T_a$ from the LST. To overcome this limitation, we employed the RF method to estimate daily Tsa on a global scale from 2000 to 2020, incorporating atmospheric and surface factors. Subsequently,

we utilized two LSTM models to generate global daily estimates of H for the same period, based on the RF-estimated Tsa and additional GLASS products. The performance of both RF and LSTM models is comprehensively assessed in Section 4, including benchmarking against various datasets and methodologies under diverse conditions. For contextual comparison, we determined the global average land surface H to be 35.29±0.71 Wm$^{-2}$ over the 2000–2020 period, surpassing the previously reported

estimates of 27 Wm$^{-2}$ by Trenberth et al. (2009) and 32 Wm$^{-2}$ by Jung et al. (2019) , and aligning closely with the 36-40 Wm$^{-2}$ range reported by Siemann et al.(2018). Despite these advancements, certain aspects still require discussion, particularly regarding the optimal selection of input data for estimating Tsa and the application of accurate Tsa.

    Existing research indicates that Tsa is affected by a blend of atmospheric and surface factors (Feng

and Zou, 2019). Theoretically, when the spatial resolution of terrain is finer than 5 km, it can modify the distribution of DSR and DLW reaching the land surface (Wang et al., 2004; Liang et al., 2024), which in turn influences the distribution of LST. Variations in LST, driven by differences in terrain characteristics and land cover types, can warm the atmosphere, altering atmospheric conditions and consequently affecting radiation variation. In this study, we utilized terrain, vegetation, and radiation-related variables

to estimate daily Tsa on a global scale. The relative importance of each variable within the RF model was quantified and ranked, with the findings detailed in Fig. 20. Among the variables analyzed, the NDVI, as a key vegetation parameter, exhibited the highest relative importance score of 25.2%. This underscores its pivotal role in estimating Tsa. Subsequent contributors included slope, LST, elevation, and DSR, with respective importance scores of 16.38%, 15.17%, 12.96%, and 11.16%. These findings suggest that both

terrain and radiation-related variables are integral to accurately estimating Tsa. Notably, slope and elevation were more critical than the other terrain-related variable, aspect, which accounted for 7.37%. Similarly, LST and DSR proved to be more impactful than DLW, which held a contribution of 4.98%.



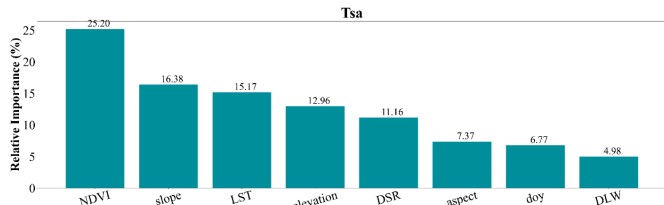

**Figure 20.** The relative importance (%) of each variable in RF model.

Moreover, Tsa is pivotal in influencing various land processes, boundary layer dynamics, weather forecasting, climate studies, and atmospheric profile retrievals. Its critical role extends to understanding atmospheric circulation, weather patterns, agricultural productivity, and ecological systems (Ghent et al., 2015; Zhang et al., 2014a). Previous studies have highlighted Tsa's significant influence on summer precipitation in the middle and lower Yangtze River (Liu et al., 2009; Zhou and Huang, 2006), and its

application in assessing soil desertification (Ai and Guo, 2003). Moreover, Tsa is instrumental in reflecting various crop development stages, such as seed germination, seedling emergence, and photosynthesis, and affects soil microbial activity and the prevalence of crop diseases and pests (Gu et al., 2012). Additionally, it serves as a crucial parameter in process-based Earth system models, indicating the intensity of land-atmosphere interactions, energy fluxes, and driving key ecological and biophysical

processes (Lensky et al., 2018; Qiang et al., 2011). The estimation of Tsa in this study further facilitates the accurate derivation of Ta values. We estimated daily Ta globally by subtracting the estimated Tsa from the GLASS LST product. For validation, we compared our Ta estimates with those from GLASS Ta using the same set of validation samples (No. of samples = 84,771. As depicted in Fig.21, the estimation Ta achieved an RMSE of 2.621 K, a MAE of 1.971 K, and an $R^2$ of 0.95, demonstrating

competitive accuracy compared to GLASS Ta (RMSE = 2.307 K, MAE = 1.692 K, $R^2$ = 0.96). These results underscore the critical role of Tsa in a wide range of environmental and agricultural applications, highlighting its significant potential for global Ta estimation and further validating the accuracy of the Tsa model.



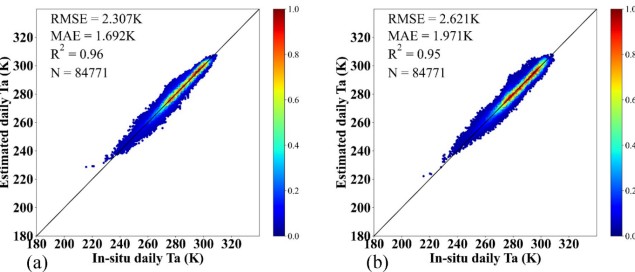

**Figure 21.** The direct validation result of Ta estimated by (a) GLASS Ta and (b) derived from GLASS LST and the estimated Tsa against in-situ measurements.

On the other hand, we investigated whether the estimated daily Tsa could enhance the accuracy of daily H values obtained from the physical model mentioned in the Introduction, compared to other data sources typically employed in existing research. To evaluate the uncertainty introduced by varying Tsa data sources, we calculated daily H using the temperature-derived method in Eq. (10):

$$H = \rho C_P (T_0 - T_a)/r_{ah} \tag{10}$$

$$r_{ah} = \frac{1}{ku^*} [\ln(\frac{(z_m-d)}{z_{om}}) - \Psi(h) + \ln(\frac{z_{om}}{z_{oh}})] \tag{10a}$$

Where $\rho$ (kg/m³) is the air density, $C_P$ (J/kg/K) is the specific heat capacity of air at constant pressure (1013), $r_{ah}$ is the aerodynamic resistance to heat transfer, $z_{om}$ (m) is roughness length for momentum transport, k is the von Karman's constant (0.41), u* is friction velocity, d is zero plane displacement height, $z_m$ (m) is the reference height, $z_{oh}$ is the roughness length for heat and related to the aerodynamic parameter $KB^{-1}$ and $z_{om}$ ($KB^{-1} = \ln(z_{om}/z_{oh})$), $\Psi(h)$ represent the stability correction functions for heat, $T_0 - T_a$ represents the Tsa and data were obtained from GLASS with 1 km resolution, estimated 1 km Tsa using a random forest (RF) model, and in-situ measurements.

Table 8 presents the results of daily H calculated from physical model by using different data sources. A total of 3,391 independent validation samples were acquired. Note that the uncertainty associated with $r_{ah}$ and $\rho C_P$ were not addressed in this study. Overall, using GLASS and estimated Tsa resulted in uncertainties of 13.5% and 5.3%, respectively, with RMSEs of 58.28 Wm⁻² and 54.08 Wm⁻², compared to Tsa from in-situ measurements (RMSE = 51.35 Wm⁻²). Additionally, the uncertainty varied across different land cover types, as shown in Table 8. Utilizing Tsa from GLASS and estimated Tsa, uncertainty ranged from 6.01% to 23.1%, with the highest and lowest uncertainties observed in GLASS for FOR and SAV, yielding RMSEs of 65.72 Wm⁻² and 59.45 Wm⁻², respectively. However, for certain land cover





types such as CRO and GRA, lower RMSEs were noted when employing Tsa from GLASS and estimated

Tsa compared to in-situ measurements, specifically. Specially, the RMSEs were 35.62 Wm⁻² and 46.2

Wm⁻² for CRO, and 46.14 Wm⁻² and 45.76 Wm⁻² for GRA. Moreover, across all five land cover types,

RMSE values consistently exceeded 35 Wm⁻² when utilizing different Tsa data sources. This could be

due to the fact that the uncertainty of parameterized method in getting $r_{ah}$ was not accounted for.

Therefore, accurately estimating $r_{ah}$ is curial in physical model and the machine learning method used

in this study effectively mitigates this issue after our experiments.

**Table 8.** The RMSE values of daily H calculated from physical model across five land cover types using
the Tsa obtained from GLASS, Estimated Tsa and in-situ measurements.

| Land cover | Data source of Tsa | | | No. of samples |
|---|---|---|---|---|
| | GLASS | Estimated Tsa | sites | |
| CRO | 35.62 | 46.2 | 46.25 | 385 |
| FOR | 65.72 | 57.46 | 53.4 | 1894 |
| GRA | 46.14 | 45.76 | 47.16 | 648 |
| SAV | 59.45 | 62.04 | 56.08 | 369 |
| SHR | 41.44 | 26.35 | 34.95 | 95 |

**6 Data availability**

The daily mean values for the first three days of each year can be freely downloaded from

https://doi.org/10.5281/zenodo.14986255 (Liang et al., 2025), and the complete products will be

available to the public at www.glass.hku.hk as soon as the manuscript is accepted.

**7 Conclusions**

To address the shortage of high-resolution and accurate data on daily land surface H, we

employed LSTM deep learning model to produce a global daily H dataset at a resolution of 1 km for the

years 2000–2020. Additionally, due to the unavailability of ABD data during the polar night, we

developed two LSTM model: one that utilizes ABD data and another that does not. Recognizing that Tsa

is a crucial driver of H and that significant uncertainty arises from the method of subtracting $T_a$ from the

LST, we introduced RF-based refined Tsa values to enhance the accuracy of H, in conjunction with five

other GLASS products, including ABD, FVC, DLW, ET and Rn. The estimation process for Tsa

integrated variables related to vegetation (NDVI), terrain (slope, aspect, and elevation), and radiation

(DSR and DLW). Validation against ground measurements demonstrated that this process for obtaining

H is more effective than other methods and products. It successfully addressed the underestimation of high H values and the overestimation of low H values, potentially due to the incorporation of time series information. When compared to the sole satellite-based H product, FLUXCOM, this method achieved the lowest RMSE of 24.5 $Wm^{-2}$ and MAE of 18.14 $Wm^{-2}$, while FLUXCOM exhibited an RMSE of

29.21 $Wm^{-2}$ and MAE of 21.82 $Wm^{-2}$. Furthermore, it demonstrated significant improvements over three other reanalysis products, with RMSE reductions of 9.351, 4.573, and 5.497 $Wm^{-2}$ and MAE reductions of 6.996, 3.562, 4.34 $Wm^{-2}$ for MERRA2, ERA5, and ERA5-Land, respectively. Several conclusions can be drawn based on the results of this study: (1) H variation exhibits clear seasonal patterns akin to those of Tsa. (2) The estimated H offer more detailed insights into heterogeneous surfaces. (3) The RF-based

refined Tsa demonstrated commendable and more robust performance. (4) Terrain significantly influences Tsa estimation, with slope being the most crucial terrain-related factor. (5) The uncertainty in the physical model was 13.5% using GLASS and 5.3% with estimated Tsa.

Overall, the daily H estimates derived from the LSTM method have demonstrated accuracy following extensive validation across diverse conditions and various products. However, significant uncertainties

persist in the South Pole region (latitude greater than 70°S) due to data scarcity, and efforts are being made to enhance the performance in these areas.

**Author contributions**

HL and SL contributed to the design of this study and developed the overall methodology. HL carried out the experiment and produced the product. SL supervised the research. HL wrote the first draft. All

the co-authors reviewed and revised the manuscript.

**Competing Interest**

The authors declare that they have no conflict of interest.

**Acknowledgements**

This work was supported by the National Natural Science Foundation of China (Grant number:

42090011). The authors gratefully acknowledge the ECMWF team for providing the ERA5 series product, the GLASS team for the GLASS product suite, the NASA team for MERRA2 and FLUXCOM





team. We thank the various networks/programs for providing in-situ measurements, including ARM, AsiaFlux, BSRN, IMAU, Lathuile (including FLUXNET and AmeriFlux), PROMICE, SURFRAD and TPDC. We also acknowledge data support from the "National Earth System Science Data Center, National Science & Technology Infrastructure of China" (http://www.geodata.cn). During the preparation of this manuscript, the authors utilized ChatGPT to enhance the language clarity and readability of the text. All content generated by the AI tool was rigorously reviewed and edited by the authors to ensure scientific accuracy and adherence to the original research intent.

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
