# Peer review of "Generation of global 1 km daily land surface - air temperature difference and sensible heat flux products from 2000 to 2020"

_Earth System Science Data, 2025_

## Author Comment (AC1)

**General comments:**

Liang et al. describe the development of a global 1km data product for land surface – air temperature differences and sensible heat fluxes. In principle such data-driven products are very valuable for the community. However, I have several methodological concerns, primarily related to the validation approach and with respect to variable selection as input to the models.

Re: Thank you very much for your comments. All comments have been well addressed one by one and provided in the follows.

**Major points**

**Point 1:**

The cross-validation strategy chosen by the authors is not adequate and yields overoptimistic results. It is absolutely compulsory to stratify train and test data by sites and not (only) by time. This is simply because data from one site are not independent and the objective of the study is to estimate at unmeasured locations. This needs to be done correctly.

Response1: Thank you for this valuable comment. There are seventeen sites in the independent validation set that were not included in the training data. The model's performance on these sites is acceptable, with an RMSE of 27.53 Wm-2, a MAE of 20 Wm-2 and an R2 of 0.43. For clarify, we have revised Fig. 8 and the corresponding text have been already added in lines 516-522 in Section 4.2 in the revised manuscript:

- 515 Fig.8a. The overall validation accuracy was deemed satisfactory, with an RMSE of 25.54 Wm-2, MAE of 18.649 Wm-2 and R2 of 0.54. To evaluate the model's ability to predict data from sites not included in the training set, we split all independent validation sample into those originating from sites used in training and those from "unseen" sites. The corresponding accuracies are presented in Fig. 8b, where the green scatter points represent samples from sites not included in the training process. The model performed
- 520 reasonably well on the "unseen" sites, with an RMSE of 27.53 Wm-2, a MAE of 20 Wm-2 and an R2 of 0.43. These results are only slightly lower than those obtained for the sites included in the training set, which yielded an RMSE of 25.16 Wm-2, MAE of 18.43 Wm-2 and R2 of 0.56. Furthermore, the spatial

sites (green scatter points). The values were obtained by replacing the results for areas with missing ABD in mod1 with those from mod2.

Moreover, H exhibits clear seasonal variations throughout the year. After stringent quality control, the daily in-situ measurements of H show substantial data gaps. To ensure the temporal continuity and completeness of the training data used in the LSTM model, we selected monthly datasets with less than 10% missing values for the training set. The remaining data were allocated to an independent validation set. The corresponding text was in lines 194-198 as "Due to significant gaps in the daily in-situ measurements of H after stringent quality control, a distinct strategy was implemented to segregate the samples for H and Tsa. For H, the methodology involved selecting monthly datasets with fewer than 10% missing values for the training set, while the rest were allocated to an independent validation set for evaluating model performance.". To maintain the representativeness of the independent validation set.

**Point 2:**

The authors chose Rn and ET as input to the model to predict H. In my opinion this is hard to justify as H=Rn-LE-G and predicting ET is a similar problem as predicting H. I would find it

conceptually more appealing if input variables are close to observations and not already derived products with additional layers of uncertainty.

Response2: Thank you for your thoughtful comment. We understand the concern regarding the use of derived variables such as Rn and ET as predictors for H, especially given the physical relationship H = Rn - LE - G.

Previous studies have shown that the variability of H is also influence by aerodynamic factors such as aerodynamic resistance which derived by wind speed. The corresponding text in lines 82-83 as "H estimation has traditionally relied on temperature-derived one-source and two-source models, incorporating ground-based observations of temperature and wind fields." and lines 89-90 as "Both models face common challenges in calculating aerodynamic resistance ( $r_{ah}$ ) due to the complexities of Monin-Obukhov similarity theory (Monin and Obukhov, 1954; Brutsaert, 2013)". However, such variables are often unavailable at the spatial resolution required for this study. In preliminary experiments, we tested the inclusion of wind speed from the MERRA-2 reanalysis dataset. However, due to its coarse spatial resolution and relatively large uncertainties, incorporating wind speed resulted in reduced model performance. In contrast, using Rn and ET—although both are derived products—provided more spatially consistent information and led to better model performance in our experiment. These variables effectively integrate various surface and atmospheric processes, thus offering informative signals for estimating H at regional to global scales. The corresponding text have been added for clarify in lines 332-336 in Section 3.2.1 in the revised manuscript:

FVC) and six radiation-related parameters (Tsa, DLW, Rn, ABD, DSR, and ET). Note that aerodynamic factors such as aerodynamic resistance, which is primarily derived from wind speed, were not included in this study. Preliminary experiments using wind speed from reanalysis datasets showed that its inclusion decreased model accuracy, likely due to the coarse spatial resolution and associated uncertainties of the data. The significant correlations among these parameters are well-established; for instance, DSR is

In future work, we plan to incorporate higher-resolution observational datasets or improved reanalysis products to further enhance the physical interpretability and robustness of the model.

**Point 3:**

335

The authors chose slope and aspect as predictors. While it is clear that these variables are very relevant in principle, the footprint of flux towers is supposed to be restricted to reasonably flat terrain. Therefore, I cannot imagine that robust patterns wrt these terrain variables can be learned.

Response3: Thank you for your insightful comment. We agree that most flux towers are installed on relatively flat terrain to ensure the validity of flux measurements. However, due to the 1 km spatial resolution of our input data, the complex surrounding terrain within and beyond the flux footprint may still affect the surface energy and temperature dynamics in the target area, even when the tower itself is situated on relatively flat ground. Moreover, as shown in Fig. 20, slope and aspect demonstrate a strong contribution to the model, highlighting their potential relevance even for towers located in predominantly flat areas. The corresponding text have been already added in lines 762-764 in Section 5 for clarity in the revised manuscript:

760 terrain and radiation-related variables are integral to accurately estimating Tsa, Notably, slope and elevation were more critical than the other terrain-related variable, aspect, which accounted for 7.37%. Although flux towers are generally installed on relatively flat terrain to ensure measurement accuracy, the surrounding complex terrain within and beyond the flux footprint can still influence local surface energy and temperature dynamics near the flux towers. Similarly, LST and DSR proved to be more

**Point 4:**

The authors also chose day of year as predictor, which has no direct environmental meaning. I suggest to drop this or replace by e.g. potential radiation or sun angle.

Response4: Thank you for your valuable suggestion. We agree that doy does not represent a direct physical quantity; however, Tsa exhibits a seasonal cycle, and doy serves as an effective temporal indicator that helps the RF model capture this variation. To evaluate its contribution, we conducted experiment and found that removing doy from the RF model resulted in a noticeable decline in performance (RMSE increased from 1.459 K to 1.51 K, MAE from 1.071 K to 1.115 K, and  $R^2$  dropped from 0.53 to 0.50). We also tested replacing doy with the solar height angle, which yielded comparable accuracy (RMSE = 1.454 K, MAE = 1.072 K, R2 = 0.53), indicating that *doy* and solar geometry-related variables provide similar predictive value. Regarding your suggestion to use potential radiation, we appreciate the insight. In our current model, we have already included radiation-related parameters such as downward shortwave radiation (DSR) and downward longwave radiation (DLW), which provide more direct and dynamic representations of surface energy input. These results support the inclusion of doy as a simple but informative feature in the Tsa estimation. Additionally, the feature importance ranking in Fig. 20 shows that while doy ranks relatively low, it still contributes to the model's performance. The corresponding text has been added to the revised manuscript (lines 765–768) for clarification.

765 impactful than DLW, which held a contribution of 4.98%. In addition, the dov ranked as the second least important variable in our analysis. Although it does not represent a direct physical environmental variable, our experiments demonstrated that it serves as a simple yet informative seasonal indicator that helps the RF model capture temporal variations effectively.

**Point 5:**

The authors mentioned a 'circular' approach between training and testing for hyper-parameter tuning (line 300). It is absolutely forbidden to use test data for any kind of model tuning. Perhaps this is a misunderstanding. Please clarify.

Response5: Thank you for pointing this out. We apologize for the misleading wording. In our workflow, the training dataset was internally partitioned into subsets for model training and hyperparameter tuning. An independent validation set was reserved exclusively for evaluating model performance and was never involved in the training or tuning process. To avoid any misunderstanding, we have revised the term "test phase" to "parameter tuning" (line 317). In addition, we carefully reviewed the entire manuscript to eliminate similar ambiguities, and the description in Section 2.1 (lines 197–207) has been revised accordingly as follows:

10% missing values for the training set, while the rest were allocated to an independent validation set for
evaluating model performance. Linear interpolation was employed to impute missing values within the
training set, ensuring the integrity of the monthly datasets. A five-fold cross-validation was then applied,
by partitioning the data such that 80% of the months were used designated for training and the remaining
20% for testing tuning the model parameters during in each iteration. This process yielded a training set
encompassing 121,542 daily H samples and an independent validation set containing 97,982 samples. In
contrast, the Tsa analysis designated measurements from 2018 to 2019 as the independent validation set
for model evaluation, with data from preceding years allocated to the training set. Specifically, for each
site, 70% of the samples from 2000 to 2017-samples were randomly selected for the training-set, and the

remaining 30% were used for testing tuning the model parameters. As a result, the Tsa training set included 564,918 daily samples, and the independent validation set comprised 84,977 daily Tsa samples.

**Point 6:**

The authors use the Twine et al approach to correct flux tower based sensible heat fluxes by forcing energy balance closure. This is a critical assumption, which needs through discussion because the uncertainty related to energy balance correction is very large, esp. for H (see

Mauder et al 2024, AFM)

Response6: Thank you for your valuable comment. We agree that correcting for energy balance closure (EBC) using the method of Twine et al. (2000) introduces uncertainty, especially for sensible heat flux. We have added a clarifying sentence in lines 188-191 in the Section 2.1 to acknowledge the assumptions and potential uncertainties associated with this correction:

Where  $H_{cor}$  is corrected H;  $LE_{aucor}$  and  $H_{uucor}$  are uncorrected LE and H, respectively. It should be noted that this correction method relies on assumptions about the distribution of residual energy, which may

190 still have uncertainties into the corrected flux values. These uncertainties and their potential impacts are further discussed in the Discussion section of this paper.

Additionally, we have expanded the Discussion section in lines 823-835 to include recent insights from Mauder et al. (2024) to discuss the uncertainties of the correct method.

Furthermore, as the H in-situ measurements used as ground truth values in this study have undergone energy balance closure (EBC) correction, their reliability warrants thorough discussion. In this study, we

- 825 adopted the widely used method proposed by Twine et al. (2000), which redistributes the residual energy between sensible and latent heat fluxes in proportion to their original magnitudes. Although this approach has been implemented in many large-scale studies and provides a practical solution when additional constraints are lacking, recent research has underscored its limitations. Notably, Mauder et al. (2024) highlighted that EBC remains a persistent issue in FLUXNET data, with a global average energy balance
- 830 ratio of approximately 0.82, and identified unresolved processes such as mesoscale secondary circulations and unmeasured energy storage terms as major contributors to the energy gap. These uncertainties are particularly relevant for H, and their effects can propagate into downstream analyses and model training. Although the Twine method does not resolve these underlying physical mechanisms,

it remains a necessary and pragmatic compromise for enabling the use of flux tower data in surface 835 energy balance studies. ↔

**Minor points:**

**Point 7:**

I find the uncertainty estimates listed in Table 1 and referenced in the text a bit misleading as they are not comparable among the products because they were not calculated consistently

Response7: Thank you for this comment. We have added the corresponding caption in Table1 for clarify as:

- Table 1. The mainstream global product information. Note that the uncertainty estimates, coming from
- 80
   different sources (e.g., documentation and publications), serve only as general references and should not be directly compared between products.

**and text in lines70-73 as:**

generally provide long temporal coverage but tend to have coarse spatial resolution and exhibit varying

70 levels of significant uncertainty, as illustrated in Table 1. Notably, the uncertainty estimates were derived through different sources (including original documentation and associated publications), and should therefore be considered as approximate references rather than being directly comparable across products. EvenFor instance, FLUXCOM\_RS,- as the most recent and only satellite product boasting with the highest spatial resolution of 0.0833°, exhibits encounters a reported global uncertainty of 11.61% over

Additionally, the corresponding context has been modified accordingly to ensure better coherence between statements.

**Point 8:**

Model tree ensembles for FLUXCOM mentioned in table 1 is likely wrong as I suppose the authors used the ensemble product

Response8: Thank you for your helpful comment. The FLUXCOM RS and RS+METEO products were generated using nine and three machine learning methods, respectively. Therefore, we have revised the term "model tree ensembles" to "multi-model ensemble" in the manuscript to more accurately reflect the methodology used in Table1.

**Point 9:**

Line 113: sentence starting with "Therefore" seems incomplete

Response9: Thank you for pointing this out. We have revised the sentences to improve its grammatical accuracy and clarity in lines 116-120. The corresponding text have been revised as "Traditional physically-based models for estimating H are typically developed for specific areas and land surface conditions, and often require parameters that are not easily accessible (e.g. aerodynamic resistance to heat transfer, rah). As a result, these models tend to produce large uncertainties when applied to other areas. Therefore, a convenient and widely applicable method for estimating global H values is still lacking."

**Point 10:**

The choice of LSTM for estimating H is unclear - have not seen a clear comparison to RF

and the other methods (did I miss this?)

Response10: Thank you for your comment. The choice of LSTM for estimating H was motivated by the limited availability of observations for H and the need to capture its temporal dependencies. This rationale is now clearly stated in lines 125–129 of the revised manuscript:

125 improving the accuracy and spatial resolution of Tsa and H on a global scale. Considering the different characteristics of the target variables, we adopted two ML models tailored. Specifically, RF was used for Tsa estimation due to the availability of dense in-situ measurements and its robust performance in such

scenarios, whereas LSTM was applied for H estimation to better handle the limited data samples and capture temporal dependencies. Given the intricate interactions between Tsa and other land-atmosphere

130 parameters, along with the significant temporal variations of H identified through our analysis, we utilized two machine learning methods, Random Forest (RF) and long short-term memory (LSTM), to predict Tsa and H, respectively. Initially, we employed the RF method, utilizing pertinent parameters

Moreover, a detailed comparison of LSTM with other methods, including RF, DBN, and Transformer, is presented in Section 4.2 (lines 536–566).

**Point 11:**

Are the comparisons of global H values in section 5 based on exactly the same spatial domain. This matters as e.g. FLUXCOM does not cover deserts where H is particularly large.

Response11: Thanks for this valuable comment. Indeed, the spatial domains and temporal periods vary across the cited studies. While these differences preclude direct quantitative comparison, we have revised the text to explicitly specify each study's spatial coverage and time periods. The corresponding modifications in lines 740-747 are as follows:

- 740 land surface H to be 35.29±0.71 Wm-2 over the 2000–2020 period, This value is higher than the 27 W m-2 based on global land data from 2000 to 2004 reported by Trenberth et al. surpassing the previously reported estimates of 27 Wm-2 by Trenberth et al. (2009), and also exceeds the 32 Wm-2 estimated by Jung et al. (2019), which excluded barren regions, deserts, permanent snow or ice, and water bodies for the period 2000–2013. It is more consistent with the \_\_\_\_\_and aligning closely with the 36-40 Wm-2 range
- reported by Siemann et al.(2018) for global land areas between 1984 and 2007. These figures are provided for general context, as differences in spatial coverage and temporal periods across studies limit direct comparability. Despite these advancements, certain aspects still require discussion, particularly

---

## Author Comment (AC2)

**General comments:**

It is a good idea to directly estimate daily land surface-air temperature difference (Tsa) and sensible heat flux (H) using machine learning method. The manuscript explores the linkage between Tsa and its predictors and demonstrates the feasibility of establishing this linkage using machine learning. The performance of the developed product is comprehensively evaluated and superior to that of corresponding reanalysis products and satellite product. I expected to see the complete product soon.

Re: Thanks for your positive comments. The data are now publicly available for download at www.glass.hku.hk.

**Point1:**

How to remove the effects of spatial resolution mismatch between multi-source predictors on the estimated Tsa and H?

Response1: Thank you for your thoughtful comment. To address the issue of spatial resolution mismatch and ensure spatial consistency across datasets, we resampled all input variables to a common spatial resolution of 1 km prior to model training. Specifically, the DSR, DLW, and Rn products (originally at 0.05° resolution) and the NDVI and LAI products (originally at 250 m resolution) were resampled using the bilinear interpolation method. This approach helps to reduce potential artifacts caused by resolution differences and ensures that the predictors are spatially aligned. A detailed description of this preprocessing step has been included in lines 237-239 of Section 2.2.2 as "To maintain spatial consistency, the DSR, DLW, and Rn products, originally at a 0.05° spatial resolution, and the NDVI and LAI products, at 250 m resolution, were resampled to 1 km using the bilinear interpolation method".

Moreover, as shown in Fig. 14, the estimated H exhibits finer spatial detail compared to FLUXCOM and ERA5-Land products, suggesting that the current resolution mismatch has a limited impact on the estimation results. In future work, we plan to further improve the model performance as higher-resolution datasets become available.

**Point2:**

If the estimated Tsa is used to derive the physically based model, what is the accuracy of H.

Response2: Thank you for your comment. To address this problem, we already conducted an independent validation using 3,391 samples, as presented in Table 8 of the manuscript. The results show that the use of estimated Tsa led to an RMSE of 54.08 W $\cdot$ m-2 (uncertainty of 5.3%),

which is comparable to the result obtained from in-situ Tsa (RMSE =  $51.35 \text{ W} \cdot \text{m}^{-2}$ ). This suggests that the estimated Tsa provides reliable input for deriving H through the physical model.

Furthermore, we analyzed the performance across different land cover types. The uncertainty varied depending on land cover, with some types (e.g., CRO, GRA and SHR) even showing lower RMSEs when using estimated Tsa compared to in-situ observations. We have clarified these findings in the manuscript (see lines 810-817 and Table 8) as:

- 810 to Tsa from in-situ measurements (RMSE = 51.35 Wm2). Additionally, the uncertainty varied across different land cover types, as shown in Table 8. Utilizing Tsa from GLASS and estimated Tsa, uncertainty ranged from 6.01% to 23.1%, with the highest and lowest uncertainties observed in GLASS for FOR and SAV, yielding RMSEs of 65.72 Wm2 and 59.45 Wm2, respectively. However, for certain land cover types such as CRO and GRA, lower RMSEs were noted when employing Tsa from GLASS and estimated
- Tsg compared to in-situ measurements, specifically. Specially, the RMSEs were 35.62 Wm2 and 46.2 Wm2 for CRO, and 46.14 Wm2 and 45.76 Wm2 for GRA. Moreover, across all five land cover types, RMSE values consistently exceeded 35 Wm2 when utilizing different Tsg data sources. This could be due to the fact that the uncertainty of parameterized method in getting  $r_{ah}$  was not accounted for. Therefore, accurately estimating  $r_{ah}$  is curial in physical model and the machine learning method used
- 820 in this study effectively mitigates this issue after our experiments.
  - Table 8. The RMSE values of daily H calculated from physical model across five land cover types using the Tsa obtained from GLASS, Estimated Tsa and in-situ measurements. ←

| Land cover | Data source of Tsa ← □ |                        |         | No. of complex/1 | ÷ |
|------------|------------------------------------------|------------------------|---------|------------------|---|
|            | GLASS↩                                   | Estimated Tsa ← | sites⇔  | INO. OI samples  | ÷ |
| CRO↩       | 35.62←                                   | 46.2←                  | 46.25←  | 385⇔             | ÷ |
| FOR←       | 65.72↩                                   | 57.46←                 | 53.4↩   | 1894←            | ÷ |
| GRA↩       | 46.14↩                                   | 45.76⇔                 | 47.16↩  | 648↩□            | ÷ |
| SAV↩       | 59.45↩                                   | 62.04←                 | 56.08↩□ | 369⇔             | ÷ |
| SHR←       | 41.44←                                   | 26.35⇔                 | 34.95↩  | 95<⊐             | ÷ |
|            |                                          |                        |         |                  |   |

It is worth noting, however, that the uncertainties associated with other parameters in the physical model (e.g., aerodynamic resistance) were not considered in this study, which may contribute to the remaining errors. We have acknowledged this limitation in lines 817-818 as "This could be due to the fact that the uncertainty of parameterized method in getting  $r_{ah}$  was not accounted for.". In addition, our results show that the machine learning-based approach adopted in this study helps mitigate the impact of such uncertainties and provides more accurate H estimates overall.

---

## Author Comment (AC3)

**General comments:**

This study presents two valuable global datasets spanning 2000-2020: land surface sensible heat flux (H) and land surface-air temperature difference (Tsa), generated using data-driven approaches. Recognizing that Tsa is the primary driver of H, the authors first refine the estimation of Tsa and subsequently employ it to predict H. Overall, the article is well-structured, with detailed explanations of the algorithm design, variable selection, and comparative analyses against existing products across multiple scales. The generated datasets hold significant value for global energy balance studies.

Re: Appreciations to your positive comments. All the comments have been replied one by one as following.

**Point1:**

A notable limitation is the absence of data for 2021–2024.

Response1: Thank you for your comment. The absence of data beyond 2020 is primarily due to the limitation of one key input variable, LST, which is currently available only up to 2020. Other auxiliary datasets have been updated to 2022 or 2023. Once the LST data and other inputs are further updated, we will update our product accordingly.

**Additionally, a few minor issues should be addressed:**

**Point2:**

Table 3: the last row, there is a spelling mistake on "Leaf area index"; keep the initial letter case consistent for variable names in the first column of the table.

Response2: Thank you for pointing this out. We have corrected the spelling of "Leaf area index" in the last row of Table 3 and ensured that the initial letter case is consistent for all variable names in the first column.

**Point3:**

In Section 2.2.2, the GLASS product provides ET and FVC data at an 8-day temporal resolution, whereas the model requires daily input. Please clarify how this temporal discrepancy was addressed (e.g., through interpolation, or another method). A detailed explanation should be included for reproducibility.

Response3: Thank you for your valuable comment. To address the temporal resolution discrepancy, we applied linear interpolation to convert the 8-day GLASS ET, FVC and ABD data to daily values, ensuring temporal consistency with the model inputs. This interpolation was performed for each pixel across the time series. We have added a detailed description of this procedure in lines 240-242 in Section 2.2.2 as "....., and the 8-day composite datasets were linearly interpolated to daily values to maintain temporal alignment with the model inputs." to enhance clarity and reproducibility.

---

## Author Comment (AC4)

**General comments:**

Considering that accurate estimation of Tsa is a fundamental prerequisite for accurately estimating H, the authors first employed a RF to estimate Tsa, followed by the estimation of H based on LSTM, ultimately producing a high-resolution dataset. This work is of significant value for studies on climate change and land–atmosphere interactions. However, the Methods section is overly detailed and somewhat cumbersome, and the Results section lacks clear organization. Therefore, the manuscript still requires further improvement before it can be considered for publication. I would like to offer the following suggestions, which I hope will useful for the authors.

Re: Thanks for your comments and suggestions. All the comments have been replied one by one as following.

**Introduction**

**Point1:**

It is recommended to consider changing the notation from " $T_{sa}$ " to " $T_{s-a}$ ".

Response1: Thank you for the suggestion. We have replaced "Tsa" with "Ts–a" throughout the manuscript as recommended, as it provides greater clarity and precision in terminology.

**Point2:**

The authors emphasize the importance of Ts–Ta in estimating sensible heat flux (H), which is understandable. However, since the H product has already been produced, it remains unclear why there is still a need to derive or produce Ts–Ta separately. The manuscript does not sufficiently justify the necessity of generating Ts–Ta as an independent product, especially given that H has already been estimated. A clearer explanation of the added value or specific applications of the Ts–Ta product is needed to support its relevance.

Response2: Thank you for your valuable comment. We would like to clarify that in our study, the H product was estimated based on the generated Ts–a product. This has been clarified in lines 133–136: "Initially, we employed the RF method, utilizing pertinent parameters mentioned above to precisely estimate Ts-a, followed by an in-depth uncertainty analysis. Subsequently, a global H product for the period of 2000 to 2020 was generated using LSTM models, incorporating data from the Global LAnd Surface Satellite (GLASS) product suite and the estimated Ts-a."

Given that Ts–a is the primary driver of H, and that directly computing it as LST minus Ta from existing products can lead to considerable uncertainties, we first developed a dedicated Ts-a dataset with improved spatial and temporal consistency. This rationale is also explained in lines 97–99: "Since Ts-a significantly influences H, its variability directly reflects in H fluctuations. Therefore, improving the accuracy of Ts–a estimation and minimizing related errors is crucial for developing a reliable, globally applicable method for H estimation," and in

lines 114–116: "Additionally, estimating Ts-a by subtracting Ta from LST, using the same or different products, can introduce significant uncertainties (Wang et al., 2020)."

Only after ensuring the reliability of the Ts-a product did we proceed with estimating H. Thus, generating Ts-a is not redundant but a necessary and foundational step. Moreover, the Ts-a product has independent value and broad application. The corresponding points are further elaborated in the Section 5.2.

**Point3:**

The manuscript does not provide a clear explain for selecting Random Forest (RF) and Long Short-Term Memory (LSTM) as the baseline methods for predicting H and Ts–Ta. The choice of these specific models requires further justification.

Response3: Thank you for this insightful comment. The selection of RF and LSTM was based on the characteristics of the target variables and the nature of the available data. To clarify this rationale, we have revised the corresponding text in lines 126–130 as follows:

"Considering the different characteristics of the target variables, we adopted two ML models tailored. Specifically, RF was used for Ts-a estimation due to the availability of dense in-situ measurements and its robust performance in such scenarios, whereas LSTM was applied for H estimation to better handle the limited data samples and capture temporal dependencies.".

**Methods**

**Point4:**

The methodology section is overly detailed. It is recommended to streamline the description to enhance clarity and readability.

Response4: Thank you for your suggestion. We have streamlined the methodology section by removing overly detailed descriptions, particularly for well-established machine learning methods and statistical metrics. The revised section now focuses on key model settings and implementation details relevant to this study, thereby improving clarity and readability.

**Point5:**

It is recommended to introduce the data used for model development and those for comparison respectively. Presenting these two types of data separately will help improve the clarity of the manuscript.

Response5: Thank you for the constructive suggestion. We fully agree that clearly distinguishing between the datasets used for model development and those used for comparison is important for improving clarity. In fact, this distinction was already presented at the beginning of Section 2 in the original manuscript, where we stated:

"This study utilized three distinct types of data: in-situ measurements, remotely sensed products, and reanalysis datasets. In-situ measurements were employed for both model development and independent validation. Remotely sensed products, including GMTED2010 DEM and GLASS product suite, supported the modeling and generation of new Ts–a and H products, while FLUXCOM was used for comparison with H estimates. Reanalysis datasets were used for comparative analysis with H and Ts–a estimates. Detailed descriptions of each dataset are provided in the subsequent sections."

We have carefully considered your suggestion regarding restructuring, and while we understand the potential benefits of presenting the two types of data separately, we found that doing so would fragment the description of certain datasets—particularly the GLASS product suite, which plays multiple roles in this study. As such, we believe that retaining the current structure, while clearly summarizing the data roles in both the main text and Table 3, offers a balanced solution that preserves both clarity and coherence.

**Result**

**Point6:**

The superiority of the Tsa dataset is primarily illustrated through comparisons with other available products. While this approach is valuable, it is recommended to include a more thorough and explicit discussion of the advantages of the generated Tsa data.

Response6: Thank you for your insightful suggestion. We agree that a clear discussion of the advantages of the generated Ts–a product is important. In the current manuscript, we have already provided a detailed evaluation of our Ts–a dataset through comprehensive comparisons with GLASS and ERA5-Land products. These comparisons, presented in Figures 6–7 and discussed in lines 476–517, include performance under various conditions such as elevation, slope, NDVI, and land cover types. Our results consistently demonstrate that the estimated Ts–a using the RF model outperforms the existing products across nearly all scenarios, particularly in complex terrain and heterogeneous land surfaces.

**Point7:**

Estimating Tsa as an intermediate step before deriving H may provide more reasonable than directly estimating H. A comparative analysis of these two approaches would be valuable in highlighting their respective strengths and limitations.

Response7: Thank you for your valuable suggestion. In the preliminary phase of our study, we tested two alternative approaches: one involved directly subtracting Ta from LST (derived from the GLASS product) to obtain Ts-a as the model input, while the other used LST alone for direct H estimation. The results showed that neither of these methods provided better predictions compared to our current approach. Therefore, we believe using Ts-a as an intermediate step for estimation better captures the close physical relationship between Ts-a and H and reduces the errors that can arise when directly estimating H. Furthermore, Ts-a itself

has practical value illustrated in section 5.2. The preliminary testing of these alternative approaches was not included in a detailed comparison within the main text, as our primary focus was on introducing the new H dataset and highlighting its generation methodology. However, we fully recognize the value of this comparative analysis, and we plan to explore it further in future research to better assess the strengths and limitations of different direct estimation approaches.

**Point8:**

The title of section 4.1 is "Uncertainty quantification of Tsa model", however, this section mainly compared product Tsa with GLASS and ERA5-land.

Response8: Thank you for pointing this out. To better reflect the actual focus of the section, we have revised the title to "Evaluation of estimated Ts–a" accordingly.

**Point9:**

P245: Please remove the validation method to Method section.

Response9: Thanks for your suggestion. Considering the characteristics of the target variables and the nature of the available data, we initially selected LSTM for predicting the temporally varying H variable. To demonstrate the superiority of this approach, we compared it in Section 4.2 with three other representative models: the generative model DBN, the tree-based model RF, and the classic time-series model Transformer. Therefore, we have retained the current structure, which we believe best represents the strengths of our model.

**Point10:**

It is noted that Figure 7 is missing tick marks for the  $R^2$  axis. Including these would enhance the figure's readability and allow for better quantitative comparison.

Response10: Thank you for your helpful comment. We have added tick marks to the  $R^2$  axis in Figure 7 as suggested to improve readability and facilitate quantitative comparison.

**Point11:**

Although comparing with other daily H datasets is valuable, the discussion spans eight paragraphs, which may overwhelm the reader. Condensing this section while retaining the key findings would improve readability and better highlight the strengths of the proposed dataset.

Response11: Thank you for your suggestion. While we understand your concern about the length of the discussion, we believe that a detailed comparison is essential for a thorough evaluation of our product. Our results first present the global spatial distribution of the products, highlighting significant spatial details. We then provide a site-level analysis, distinguishing

between remote sensing and reanalysis products. Each product is further analyzed across various conditions, which we consider necessary for a comprehensive assessment. Moreover, given the current absence of comparative studies on sensible heat flux products in this context, a detailed comparison is crucial for understanding both the global distribution and the performance and limitations. Therefore, we have decided to retain the current level of detail in this part to ensure a complete and robust evaluation.

**Discussion**

**Point12:**

To enhance readability and guide the reader through the key arguments, the Discussion section would benefit from the inclusion of subheadings that reflect its main themes.

Response12: Thank you for your suggestion. We have added subheadings in the Discussion section to improve readability and guide the reader through the key themes.

**Point13:**

While the analysis of variable importance in estimating Tsa is useful, the primary objective of the study is the accurate estimation of H. Therefore, investigating the contribution of each variable to the estimation of H would be more directly aligned with the study's goals and would provide greater practical insight.

Response13: Thank you for your thoughtful suggestion. We agree that examining the contribution of each variable to H estimation would offer valuable insights. In this study, we did not perform such an analysis because our input variables were selected based on established literature (e.g., (Wulfmeyer et al. 2022)) where their importance has already been thoroughly examined. Additionally, we also conducted correlation analyses during feature selection (Section 3.2.1), which provide indirect evidence of each variable's influence. Moreover, since H was estimated using an LSTM model, it is not straightforward to interpret individual variable contributions as in models like Random Forest. In contrast, Ts–a is influenced by a complex interplay of atmospheric and surface factors, and the contribution of individual variables to its estimation has been less explored in previous studies. Therefore, we placed greater emphasis on discussing the drivers of Ts–a in the manuscript, as it serves as the key variable for deriving H.

Wulfmeyer, V., Pineda, J.M.V., Otte, S., Karlbauer, M., Butz, M.V., Lee, T.R., & Rajtschan, V. (2022). Estimation of the Surface Fluxes for Heat and Momentum in Unstable Conditions with Machine Learning and Similarity Approaches for the LAFE Data Set. Boundary-Layer Meteorology, 186, 337-371

**Point14:**

The current manuscript outlines the application scenarios of Tsa; however, since H is the final product of interest, its application scenarios should also be discussed. This will better demonstrate its scientific and practical significance.

Response14: Thank you for your valuable suggestion. We have already outlined the potential applications of H in the first paragraph of the Introduction, where we highlight its critical role in global energy flows, land-atmosphere interactions, and its significant impact on climate and weather systems. This provides a comprehensive understanding of its applications and demonstrates the significance of our dataset from the outset. We believe this approach ensures a clear and focused presentation of the study's contributions, which is why we have not included this information in the Discussion section.

Regarding Ts-a, as it serves as an intermediate product, its application scenarios and significance require further discussion. Therefore, we have placed the application scenarios of Ts-a in the Discussion section.

**Conclusions**

**Point15:**

Please consider refining and condensing the description of both the methods and the main conclusions. This will help to emphasize the mainly point of the manuscript.

Response15: Thank you for your suggestion. We have revised the manuscript to condense and refine the description of both the methods and the main conclusions.